# Bacterial lifestyle switch in response to algal metabolites

Noa Barak-Gavish[1†], Bareket Dassa[2], Constanze Kuhlisch[1], Inbal Nussbaum[1], Alexander Brandis[2], Gili Rosenberg[3], Roi Avraham[3], Assaf Vardi[1]*

[1]Department of Plant and Environmental Sciences, Weizmann Institute of Science, Rehovot, Israel; [2]Life Sciences Core Facilities, Weizmann Institute of Science, Rehovot, Israel; [3]Department of Biological Regulation, Weizmann Institute of Science, Rehovot, Israel

**Abstract** Unicellular algae, termed phytoplankton, greatly impact the marine environment by serving as the basis of marine food webs and by playing central roles in the biogeochemical cycling of elements. The interactions between phytoplankton and heterotrophic bacteria affect the fitness of both partners. It is becoming increasingly recognized that metabolic exchange determines the nature of such interactions, but the underlying molecular mechanisms remain underexplored. Here, we investigated the molecular and metabolic basis for the bacterial lifestyle switch, from coexistence to pathogenicity, in *Sulfitobacter* D7 during its interaction with *Emiliania huxleyi*, a cosmopolitan bloom-forming phytoplankter. To unravel the bacterial lifestyle switch, we analyzed bacterial transcriptomes in response to exudates derived from algae in exponential growth and stationary phase, which supported the *Sulfitobacter* D7 coexistence and pathogenicity lifestyles, respectively. In pathogenic mode, *Sulfitobacter* D7 upregulated flagellar motility and diverse transport systems, presumably to maximize assimilation of *E. huxleyi*-derived metabolites released by algal cells upon cell death. Algal dimethylsulfoniopropionate (DMSP) was a pivotal signaling molecule that mediated the transition between the lifestyles, supporting our previous findings. However, the coexisting and pathogenic lifestyles were evident only in the presence of additional algal metabolites. Specifically, we discovered that algae-produced benzoate promoted the growth of *Sulfitobacter* D7 and hindered the DMSP-induced lifestyle switch to pathogenicity, demonstrating that benzoate is important for maintaining the coexistence of algae and bacteria. We propose that bacteria can sense the physiological state of the algal host through changes in the metabolic composition, which will determine the bacterial lifestyle during interaction.

*For correspondence:
assaf.vardi@weizmann.ac.il

Present address: †Institute of Microbiology, ETH Zurich, Zurich, Switzerland

Competing interest: The authors declare that no competing interests exist.

## Editor's evaluation

This paper presents important new findings on the mechanism by which a bacterial species switches between co-existence with a bloom-forming phytoplankter to being pathogenic. Specifically, the study identifies algal dimethylsulfoniopropionate (DMSP) as a key chemical component that triggers the bacterial switch. The results are convincing and will be of interesting to scientists interested in inter-kingdom microbial communication including microbiologists and biologists working with algae.

## Introduction

Half of Earth's photosynthesis takes place in the marine environment by phytoplankton – photosynthetic single-celled algae (*Field et al., 1998*). Phytoplankton have great ecological importance by forming the basis of marine food webs and influencing biogeochemical cycles. Therefore, the biotic interactions phytoplankton engage in, and the metabolic exchange that governs them, have immense

impacts on large-scale biogeochemical processes. Phytoplankton are a main source of organic matter in the marine environment, thus fueling the growth and functioning of heterotrophic bacteria that interact with them through chemical exchange (*Cirri and Pohnert, 2019*; *Seymour et al., 2017*). Chemical communication takes place in the phycosphere – the diffusive boundary layer that surrounds algal cells, where molecules can accumulate to high concentrations (*Bell and Mitchell, 1972*; *Seymour et al., 2017*). Studies on algae-bacteria interactions revealed that the partners often exchange growth substrates (*Landa et al., 2017*; *Segev et al., 2016*), essential vitamins and nutrients (*Amin et al., 2009*; *Croft et al., 2005*; *Wang et al., 2014*), and infochemicals (molecules that convey information) (*Amin et al., 2015*; *Barak-Gavish et al., 2018*; *Pohnert et al., 2007*; *Seyedsayamdost et al., 2011*). Bacteria have developed mechanisms of foraging for phytoplankton cells, such as motility and chemotaxis, and cell-surface attachment mechanisms to maintain close association within the phycosphere (*Fei et al., 2020*; *Furusawa et al., 2003*; *Li et al., 2016*; *Mayali et al., 2008*; *Miller and Belas, 2006*; *Slightom and Buchan, 2009*; *Sonnenschein et al., 2012*; *Stocker and Seymour, 2012*).

Marine bacteria from the *Rhodobacteraceae* family, often termed the Roseobacter group (*Simon et al., 2017*), are found to be associated with phytoplankton (*Alavi et al., 2001*; *Amin et al., 2012*; *Behringer et al., 2018*; *Buchan et al., 2014*; *Geng and Belas, 2010*; *González and Moran, 1997*; *Rink et al., 2007*; *Vincent et al., 2021*). They are metabolically versatile and specialize on algae-derived substrates that promote interactions with phytoplankton (*Newton et al., 2010*). The organosulfur molecule dimethylsulfoniopropionate (DMSP), produced by many phytoplankton species (*Keller, 1989*), is especially known to mediate Roseobacter-phytoplankton interactions by serving as a carbon and sulfur source, as a chemotaxis cue, and as an infochemical for the presence of algae (*Amin et al., 2015*; *Barak-Gavish et al., 2018*; *Bürgmann et al., 2007*; *Landa et al., 2017*; *Miller et al., 2004*; *Miller and Belas, 2004*; *Seymour et al., 2010*; *Sule and Belas, 2013*). In mutualistic interactions, algae provide organic matter such as sugars, amino acids, sulfonates, and polyamines for bacterial growth. In exchange, Roseobacters produce essential B-vitamins and growth promoting factors such as indole-3-acetic acid (*Amin et al., 2015*; *Cooper et al., 2019*; *Durham et al., 2015*; *Landa et al., 2017*; *Wagner-Döbler et al., 2010*). In recent years, cumulating studies that investigated the interactions of phytoplankton and bacteria in co-cultures revealed that some Roseobacters display a lifestyle switch from mutualism to pathogenicity toward the algae (*Barak-Gavish et al., 2018*; *Bolch et al., 2017*; *Bramucci et al., 2018*; *Mayers et al., 2016*; *Segev et al., 2016*; *Wang et al., 2014*). This occurs when the algal host reaches stationary phase and is mediated by infochemicals. For example, Roseobacters can produce potent algicidal compounds, termed roseobacticides, in response to *p*-coumaric acid, an aromatic lignin breakdown product released by aging algae (*Seyedsayamdost et al., 2011*; *Sule and Belas, 2013*). While this bacterial lifestyle switch, often termed the "Jekyll-and-Hyde" phenotype, seems to be a recurring phenomenon, knowledge about the bacterial behavior in the different modes of interaction and the regulation of such lifestyle switch are still rudimentary.

In the current study, we investigated the behavior of the Roseobacter *Sulfitobacter* D7, during interaction with *Emiliania huxleyi*, a cosmopolitan bloom-forming phytoplankter. *E. huxleyi* has a significant role in biogeochemical cycling of carbon and sulfur. It produces the climatically active gas dimethyl sulfide and its precursor DMSP, both function as infochemicals during interactions with *E. huxleyi* (*Barak-Gavish et al., 2018*; *Shemi et al., 2021*). *Sulfitobacter* sp. are associated with *E. huxleyi* in nature, and *Sulfitobacter* D7 was isolated from a natural *E. huxleyi* bloom (*Ankrah et al., 2014*; *Barak-Gavish et al., 2018*; *Ku et al., 2018*; *Vincent et al., 2021*). Therefore, this ecologically relevant model provides a tractable system to examine how metabolic exchange regulates the nature of interactions between algae and bacteria. Our previous work revealed that *Sulfitobacter* D7 displays a lifestyle switch, from coexistence to pathogenicity, during its interaction with *E. huxleyi* (*Barak-Gavish et al., 2018*). We found that algal DMSP, which usually mediates mutualistic interactions, plays a pivotal role by invoking bacterial pathogenicity (*Barak-Gavish et al., 2018*). Bacterial genes related to DMSP uptake and catabolism have been investigated in many studies (*Curson et al., 2011*; *Gao et al., 2020*; *Howard et al., 2006*; *Reisch et al., 2011*; *Sun et al., 2012*; *Todd et al., 2007*), but the regulation of DMSP-responsive genes and their interplay with bacterial lifestyle and behavior during interactions with algae is yet to be explored. We performed a transcriptomics experiment that enabled us to elucidate the bacterial response to algal infochemicals and to characterize DMSP-responsive and pathogenicity-related genes. We revealed the signaling role of DMSP that led to a systemic remodeling of *Sulfitobacter* D7 gene expression but only in the presence of additional algal

metabolites. Overall, we unraveled the transcriptional signature of the switch from coexistence to a pathogenic bacterial lifestyle during interaction with their algal host and provide insights into the ecological context of this mode of interaction.

## Results

### *E. huxleyi*-derived exudates induce remodeling of *Sulfitobacter* D7 transcriptome

The interaction between *Sulfitobacter* D7 and *E. huxleyi* displays two distinct phases (*Figure 1a*). Initially, there is a coexisting phase in which the alga grows exponentially, and the bacterium grows as well. The interaction shifts to pathogenic when the virulence of *Sulfitobacter* D7 toward *E. huxleyi* is invoked upon exposure to high concentrations of algal DMSP, which occurs when the alga reaches stationary phase or when DMSP is applied exogenously to algae in exponential growth (*Figure 1a*; *Barak-Gavish et al., 2018*). We aimed to unravel the response of *Sulfitobacter* D7 to the pathogenicity-inducing compound, DMSP, and to different alga-derived infochemicals that affect the lifestyle of the bacterium. We grew *Sulfitobacter* D7 in conditioned media (CM) derived from algal cultures at exponential and stationary phase (Exp-CM and Stat-CM, respectively), in which DMSP concentration is low and high, respectively (*Barak-Gavish et al., 2018*; *Figure 1b*, *Table 1*). This enabled us to dissect the interaction with *E. huxleyi* into its different phases, that is, Exp-CM represents the coexisting phase, and Stat-CM represents the pathogenic phase. An additional pathogenicity-inducing treatment was Exp-CM supplemented with 100 µM DMSP (herein Exp-CM + DMSP). This condition mimicked co-cultures to which we added DMSP exogenously and thus induced *Sulfitobacter* D7 pathogenicity, which led to the death of exponentially growing *E. huxleyi* (*Figure 1a*). While the concentration of DMSP measured in natural seawater is typically in the tens of nanomolar range (*Barak-Gavish et al., 2018*; *Kettle et al., 1999*), algae-derived compounds are present in higher concentrations in the phycosphere (*Seymour et al., 2017*; *Stocker, 2012*). Furthermore, the concentrations of DMSP measured in axenic stationary phase *E. huxleyi* cultures reach up to 70 µM (*Barak-Gavish et al., 2018*). We therefore argue that the 100 µM concentration used in our study is appropriate and ecologically relevant, especially in light of the high densities of the algal and bacterial cultures used in our experimental system.

In order to reveal the bacterial transcriptional response to algal exudates, we harvested bacterial cells after 24 hr of growth in the different CM treatments and performed RNAseq analysis, using a modified protocol based on *Avraham et al., 2016* (*Figure 1b*, *Table 1*). We aimed to identify pathogenicity-related genes by comparing *Sulfitobacter* D7 gene expression profiles in the pathogenicity-inducing media to the coexistence medium. We further aimed to find bacterial genes that are specifically responsive to DMSP and are not affected by other alga-derived factors. Therefore, we grew *Sulfitobacter* D7 in defined minimal medium (MM), which lacks algal metabolites, supplemented it with 100 µM DMSP (herein MM + DMSP), and examined the transcriptional response. This experimental design allowed us to expand our understanding on the bacterial response to DMSP, algal infochemicals and which of these are essential for coexistence and pathogenicity.

Among the 3803 genes in *Sulfitobacter* D7 genome, we detected the expression of 2588 genes (*Figure 1—source data 2*). Principle component analysis (PCA), based on the overall expression profile, showed a clear separation between the different CM treatments, while the MM ± DMSP samples clustered together (*Figure 1c*). Pearson correlation analyses of all samples indicated high correlation between the triplicates of each treatment, and hierarchical clustering showed that MM samples clustered separately from the CM samples (*Figure 1—figure supplement 1*). Among the CM samples, Stat-CM and Exp-CM + DMSP clustered together and had higher correlation to each other compared to Exp-CM (*Figure 1—figure supplement 1*). This suggests that the pathogenicity-inducing media elicit a set of expressed genes, distinct from that induced by the coexistence medium.

To identify the pathogenicity- and DMSP-related genes, we examined the genes that were differentially expressed (DE) in the following comparisons: Exp-CM+DMSP vs. Exp-CM, Stat-CM vs. Exp-CM, and MM +DMSP vs. MM. We defined significantly DE genes as |fold change|>2 and adjusted p-value ≤0.05. The DE genes were separated into four clusters, based on k-means clustering, and we assessed the enrichment in KEGG pathways in each cluster (*Figure 1d*). Cluster 1 contained 197 genes that were responsive to DMSP, namely DE in the Exp-CM±DMSP and the MM ±DMSP comparisons.

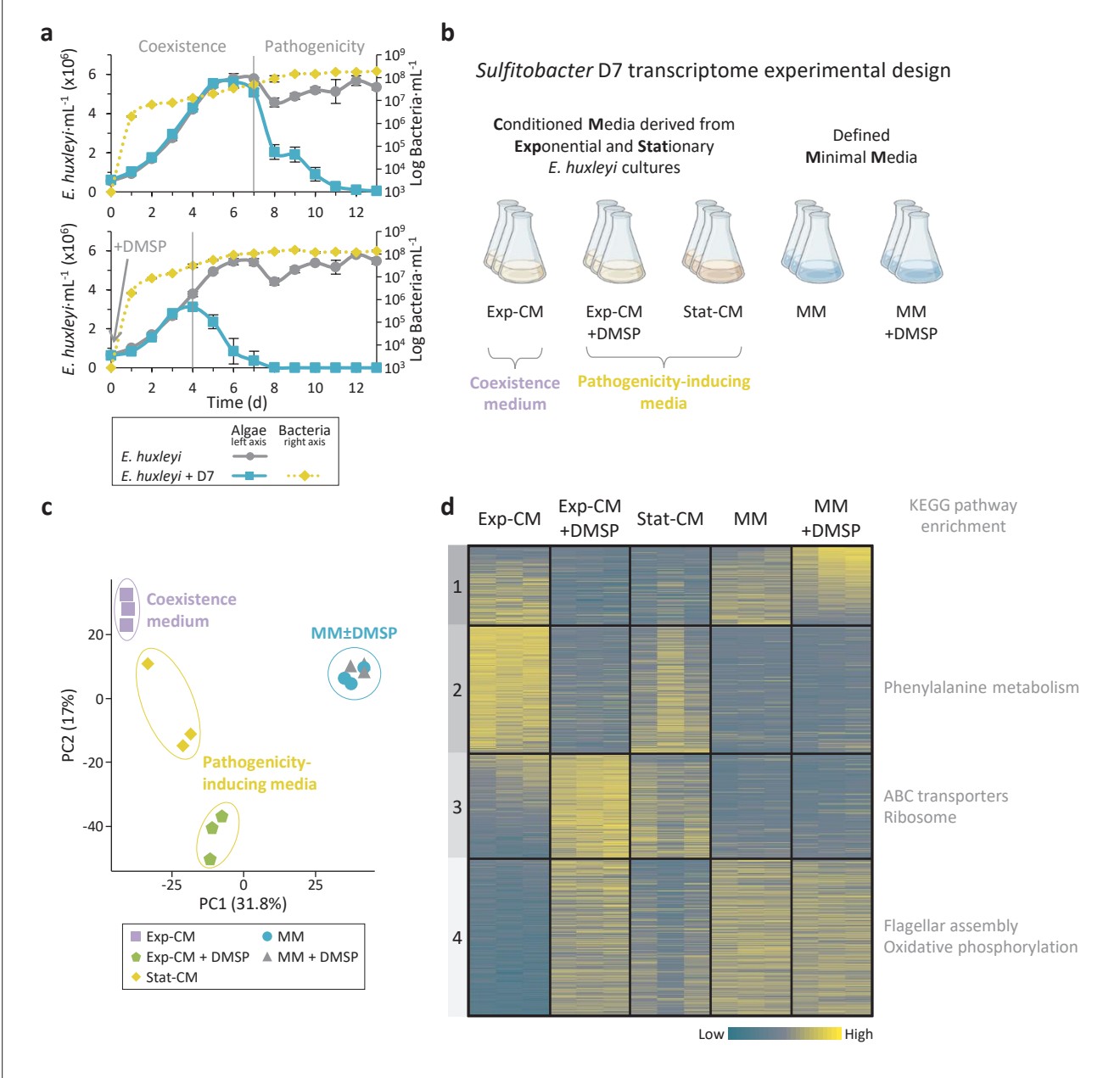

**Figure 1.** Transcriptional profiling of *Sulfitobacter* D7 in response to dimethylsulfoniopropionate (DMSP) and additional *Emiliania huxleyi* infochemicals reveals the signaling role of DMSP. (**a**) Time course of *E. huxleyi* CCMP379 and bacterial abundance (full and dashed lines, left and right axes, respectively) in algal mono-cultures or during co-culturing with *Sulfitobacter* D7. Top panel: Co-cultures display two phases with distinct bacterial lifestyles: coexistence and pathogenicity. Bottom panel: DMSP was added at day 0 to a final concentration of 100 μM. Results represent average ± SD (n=3). *Figure 1a* has been adapted from Figure 5 of *Barak-Gavish et al., 2018*. (**b**) Design of *Sulfitobacter* D7 transcriptome experiment aiming to explore gene expression profiles in response to *E. huxleyi*-derived media and in response to DMSP. Growth media consisted of conditioned media (CM) derived from *E. huxleyi* at exponential growth or stationary phase (Exp-CM and Stat-CM, respectively), and an additional treatment in which 100 μM of DMSP was added (Exp-CM + DMSP). These media differentially induce the coexistence and pathogenicity lifestyles of *Sulfitobacter* D7. In order to identify DMSP-responsive genes we inoculated *Sulfitobacter* D7 in defined minimal media (MM), lacking *E. huxleyi*-derived exudates, without and with 100 μM DMSP (MM and MM + DMSP, respectively). *Sulfitobacter* D7 was inoculated into each media and harvested for RNA profiling after 24 hr of growth. Initial conditions of the media and bacterial growth are elaborated in *Table 1*. (**c**) Principle component analysis of *Sulfitobacter* D7 detected genes in all treatments (2588 genes). Triplicates of each treatment are shown. (**d**) Heatmap of gene expression of all differentially expressed genes in the comparisons Exp-CM + DMSP vs. Exp-CM, Stat-CM vs. Exp-CM, and MM + DMSP vs. MM (1179 genes). Clusters were determined based on k-means analysis. Significant functional enrichment in each cluster, based on Kyoto encyclopedia of genes and genomes (KEGG) pathways, is denoted. Each row represents one gene, and the color intensity corresponds to the standardized expression across all samples (triplicates of each treatment are shown).

*Figure 1 continued on next page*

*Figure 1 continued*

Expression values are scaled by row. Genes in cluster 1 are ordered based on the mean expression values in the MM + DMSP treatment. Genes in cluster 2–4 are ordered based on the mean expression values in the Exp-CM treatment.

The online version of this article includes the following source data and figure supplement(s) for figure 1:

**Source data 1.** Time course of *E. huxleyi* CCMP379 and *Sulfitobacter* D7 abundances.

**Source data 2.** RNA sequencing data of *Sulfitobacter* D7 transcriptome.

**Source data 3.** *Sulfitobacter* D7 expression of genes belonging to KEGG pathways that are enriched in clusters 2–4 of the heatmap in *Figure 1d*.

**Source data 4.** Number of sequencing reads of *Sulfitobacter* D7 transcriptome experiment.

**Figure supplement 1.** Gene expression similarity between the treatments of *Sulfitobacter* D7 transcriptome.

Interestingly, the expression pattern of cluster 1 genes in response to DMSP was largely different: in Exp-CM, DMSP led to downregulation, while in MM it led to upregulation. The differential effect of DMSP in Exp-CM and MM is also visualized in the PCA and is evident by the number of DE genes in each comparison: 968 genes were DE between Exp-CM vs. Exp-CM+DMSP, while only 170 genes were DE between MM vs. MM +DMSP (*Table 2*). Since the metabolic composition of these two media was profoundly different, it suggests that the effect of DMSP signaling depends on the simultaneous perception of other metabolites. Namely, in different chemical contexts, DMSP will affect *Sulfitobacter* D7 gene expression in different ways.

Cluster 2 contained 322 genes that were highly expressed in the coexistence medium, Exp-CM, compared to the pathogenicity-inducing media, therefore, we consider it as coexistence-related. This cluster was enriched with genes related to the phenylalanine metabolism pathway, and specifically to phenylacetic acid (PAA) degradation (*Figure 1—source data 3*). PAA is a phytohormone that can potentially promote algal growth (*Cook, 2019*), and PAA metabolism in bacteria relates to production of secondary metabolites that can affect algae (*Thiel et al., 2010*; *Wang et al., 2016*). Clusters 3 and 4 contained 264 and 396 genes, respectively, that were mainly upregulated in Exp-CM +DMSP and Stat-CM compared to Exp-CM, and we consider these as pathogenicity-related clusters. Cluster 4 also contained genes that were highly expressed in MM ±DMSP. Cluster 3 was enriched with genes encoding for ABC transporters and ribosomal proteins, and cluster 4 was enriched with flagellar genes and genes related to oxidative phosphorylation (*Figure 1d*, *Figure 1—source data 3*). The enrichment in genes encoding for ribosomal proteins and an F-type ATPase, related to oxidative phosphorylation, in the pathogenicity-related clusters suggests that during the pathogenic lifestyle *Sulfitobacter* D7 may be more metabolically active than in the coexistence lifestyle (*Figure 1—source data 3*). Overall, this transcriptomics experimental setup enabled us to capture the gene expression of *Sulfitobacter* D7 in coexistence and pathogenicity modes. Moreover, it demonstrated the pivotal

**Table 1.** Conditions of the media used for *Sulfitobacter* D7 transcriptome.

| Media | *Sulfitobacter* D7 abundance at t=24 hr ($10^6 \cdot mL^{-1}$)† | DMSP concentration (µM)‡ | For conditioned media (CM)* | | |
|---|---|---|---|---|---|
| | | | Days of *E. huxleyi* growth | *E. huxleyi* abundance (cells·$mL^{-1}$) | % Dead cells‡ |
| Exp-CM | 2.6±0.1 | 3 | 5 | $1 \cdot 10^6$ | 8.5% |
| Exp-CM + DMSP | 3.7±0.3 | 100 | 5 | $1 \cdot 10^6$ | 8.5% |
| Stat-CM | 17.5±0.8 | 10.5 | 12 | $3 \cdot 10^6$ | 18% |
| MM | 7.5±0.5 | 0 | – | – | – |
| MM + DMSP | 18.6±1.3 | 100 | – | – | – |

DMSP, dimethylsulfoniopropionate; CM, conditioned media; MM, minimal media; Exp-CM, exponential CM; Stat-CM, stationary CM.

*Conditions of exponential and stationary *E. huxleyi* cultures from which Exp-CM and Stat-CM were obtained, respectively (n=1).

†Results represent average ± SD (n=3). At t=24 hr bacteria were harvested for transcriptome analysis (experimental setup, *Figure 1b*).

‡DMSP and cell death measurements are described in *Barak-Gavish et al., 2018* (n=1).

**Table 2.** Number of *Sulfitobacter* D7 differentially expressed genes in pathogenicity vs. coexistence modes and in response to DMSP.

| | Exp-CM +DMSP vs. Exp-CM | Stat-CM vs. Exp-CM | MM +DMSP vs. MM |
|---|---|---|---|
| Upregulated | 560 | 358 | 107 |
| Downregulated | 408 | 237 | 63 |
| Total | 968 | 495 | 170 |

DE genes were defined as genes with |fold change|>2 and adjusted *P*-value ≤0.05.
DMSP, dimethylsulfoniopropionate; CM, conditioned media; MM, minimal media; Exp-CM, exponential CM; Stat-CM, stationary CM.

role of DMSP in regulating gene expression, which depends also on the chemical environment. In *E. huxleyi*-derived CM, DMSP led to major changes in *Sulfitobacter* D7 transcriptome, while in MM, which lacked additional algal metabolites, DMSP had a minor effect on gene expression. We therefore suggest that the additional algal factors act in concert with DMSP and are required for the expression of coexistence- and pathogenicity-related genes.

## The pathogenic lifestyle of *Sulfitobacter* D7 includes upregulation of flagellar genes and increased motility

The enrichment in flagellar genes in cluster 4 suggests that flagellar motility may be involved in the pathogenic lifestyle of *Sulfitobacter* D7. We examined the expression of all the genes necessary for flagellar assembly, which are localized in a gene cluster in the genome of *Sulfitobacter* D7 (*Figure 2a*). Most of the flagellar genes were significantly upregulated in pathogenicity-inducing media compared to the coexistence medium (*Figure 2a*), including the genes encoding for the flagellar hook and basal body (*Figure 2a* and *Figure 2—source data 1*). The flagellar genes FliC, FliM, FlgC, FlgB, and FliI were not DE but were highly expressed in all treatments (*Figure 2—source data 1*). The genes FlhB, FliR, FlhA, and FliQ were not sufficiently detected in our analysis. Interestingly, in MM there were no significant changes in expression of flagellar genes in response to DMSP, although the overall expression was higher than in the CM samples (*Figure 2a*, *Figure 2—source data 1*).

To assess the involvement of motility in the behavioral switch of *Sulfitobacter* D7 and to validate the expression patterns of flagellar genes, we performed a bacterial motility assay in response to *E. huxleyi*-derived metabolites. We examined the colony expansion of *Sulfitobacter* D7 in semi-solid agar plates composed of Exp-CM, Exp-CM +DMSP, or Stat-CM. We first pre-conditioned *Sulfitobacter* D7 in the respective liquid CM for 24 hr, in order to induce the appropriate expression of flagellar genes. *Sulfitobacter* D7 plated on pathogenicity-inducing semi-solid media showed higher colony expansion areas than in the coexistence medium, indicating increased motility under these conditions (*Figure 2b–c*). The average colony area in semi-solid Stat-CM and Exp-CM +DMSP was 37.0 and 20.8 mm$^2$, respectively, while in Exp-CM it was only 13 mm$^2$ (*Figure 2b*). Moreover, the morphology of the colonies was different; the colony edges in Stat-CM were smeared, and there were bacterial motility extensions from the core colony, indicating bacterial migration in the semi-solid agar (*Figure 2c*). The smeared edges were also evident in Exp-CM +DMSP, but to a lesser extent. These results validated the expression patterns of flagellar genes in each CM. Interestingly, *Sulfitobacter* D7 that was pre-grown in liquid marine broth (½MB), and was therefore not pre-exposed to *E. huxleyi* infochemicals, and subsequently plated on the three semi-solid CM did not show major differences in the average colony area (*Figure 2b–c*). This strongly indicates that *Sulfitobacter* D7 grown in liquid pathogenicity-inducing media were pre-conditioned for motility by upregulating the expression of flagellar genes compared to the coexistence medium. Still, even without the priming in liquid pathogenicity-inducing media, bacterial colonies grown in liquid ½MB and plated on Stat-CM were significantly larger and showed the smeared edges morphology, implying that in semi-solid Stat-CM there was also induction of motility (*Figure 2b–c*). Taken together, high expression of flagellar genes in pathogenicity-inducing media, along with the observation that bacteria are indeed more motile in these conditions, indicates that flagella-driven motility may be involved in the pathogenic lifestyle of *Sulfitobacter* D7 during interaction with *E. huxleyi*.

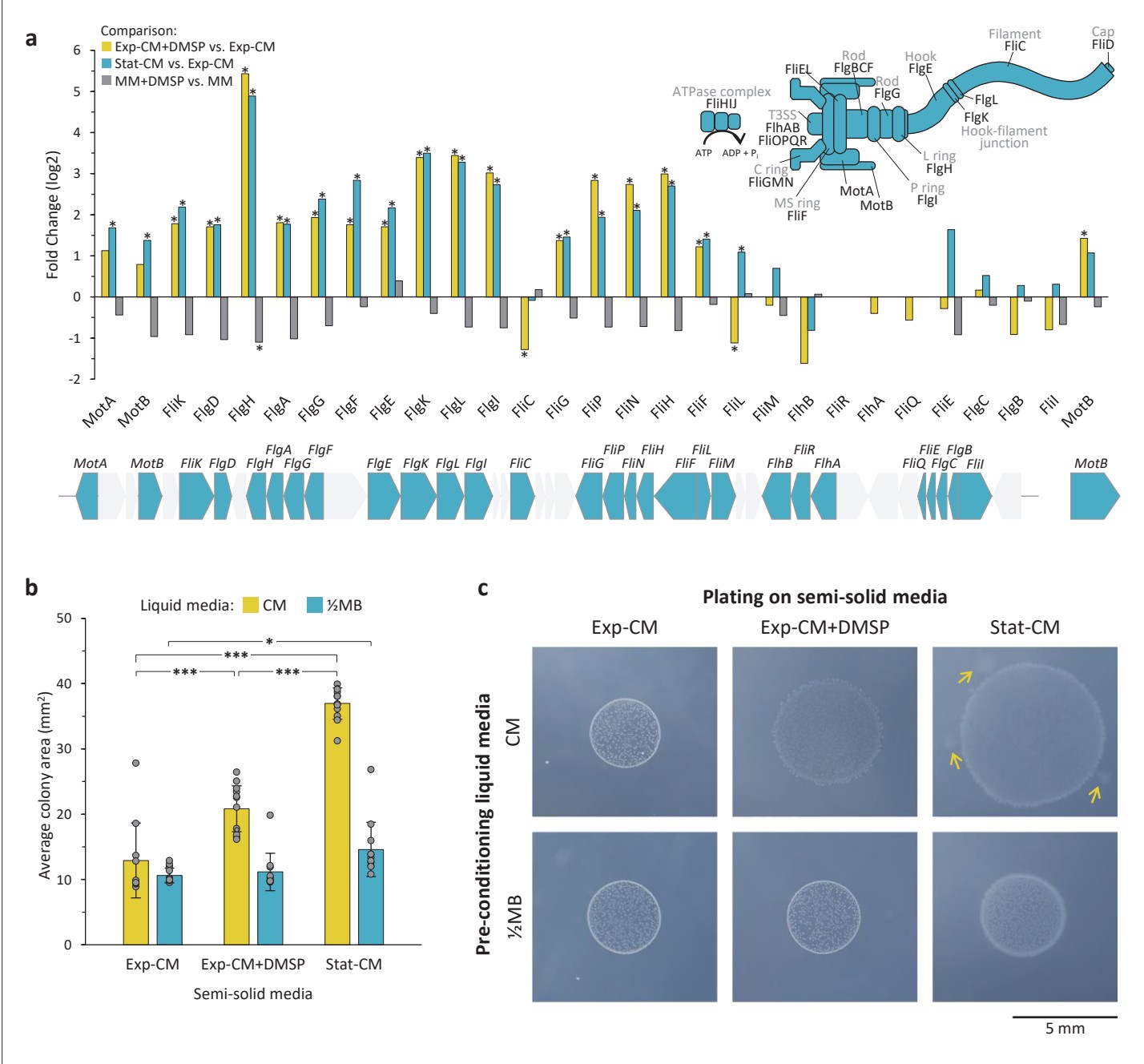

**Figure 2.** Induction of flagellar genes and increased motility in pathogenicity-inducing media. (**a**) Fold change of flagellar gene expression in the comparisons: Exp-CM +DMSP vs. Exp-CM (yellow), Stat-CM vs. Exp-CM (blue), and MM +DMSP vs. MM (gray). Genes marked with * are significantly differentially expressed. The flagellar genes are localized in a gene cluster on the *Sulfitobacter* D7 chromosome, as shown below the graph. Genes in gray are not related to the flagellum. The function of each gene is indicated in the flagellum assembly scheme on the top-right. Expression values are presented in *Figure 2—source data 1*. (**b**) Bacterial motility inferred by the colony area of *Sulfitobacter* D7 plated on semi-solid agar media. Bacteria were pre-conditioned in liquid CM (Exp-CM, Exp-CM + DMSP, and Stat-CM) for 24 hr and plated on the corresponding semi-solid CM plates (yellow bars). For control, bacteria were pre-conditioned on liquid marine broth (½MB) and plated on semi-solid CM plates (blue bars). Colony area was determined after 6 days of growth. Results represent average ± SD (n=10–12 colonies per treatment). Statistical differences were tested using two-way analysis of variance, followed by Tukey's post-hoc test. * p-value <0.05, *** p-value <0.0001. (**c**) Representative bacterial colonies from each treatment showing the difference in colony area and morphology. The arrows depict bacterial motility extensions from the core colony. The extensions were not included in the colony area measurements. DMSP, dimethylsulfoniopropionate; CM, conditioned media; MM, minimal media; Exp-CM, exponential CM; Stat-CM, stationary CM.

The online version of this article includes the following source data for figure 2:

*Figure 2 continued on next page*

*Figure 2 continued*

**Source data 1.** *Sulfitobacter* D7 expression of flagellar genes.

**Source data 2.** Bacterial motility inferred by the colony area of *Sulfitobacter* D7 plated on semi-solid agar media.

## DMSP and *E. huxleyi*-derived metabolites modulate the expression of *Sulfitobacter* D7 transport genes

The enrichment in ABC transporters in cluster 3 suggests that nutrient uptake by *Sulfitobacter* D7 is prominent during the interaction with *E. huxleyi*. We thus examined the expression of all 493 transport genes in *Sulfitobacter* D7 genome. We found that transporters for energy-rich organic compounds were expressed in CM treatments (*Figure 3*, *Figure 3—source data 1*). This includes transporters for amino acids and peptides, carbohydrates and sugars, organic sulfur and nitrogen compounds, as well as for inorganic nutrients and metals. Examination of bacterial transport genes was shown to serve as a sensitive readout for estimating which metabolites reside in the medium and are taken up by bacteria (*Ferrer-González et al., 2021*). Therefore, the elevated expression of transporters implies that the CM contained *E. huxleyi*-derived metabolites that *Sulfitobacter* D7 can benefit from during growth in CM and during the interaction with the alga. Such metabolites include branched-chain amino acids, sugars, C4 carbohydrates, and DMSP, which are known to be produced by *E. huxleyi* (*Obata et al., 2013*; *Tsuji et al., 2012*; *Figure 3*).

Numerous transport genes were DE in the pairwise comparisons of the different treatments (*Figure 3*, *Figure 3—figure supplement 1*, and *Figure 3—source data 1*). *Sulfitobacter* D7 grown in Stat-CM had a similar expression profile of transport genes to that of Exp-CM +DMSP (*Figure 3*, *Figure 3—figure supplement 1*, and *Figure 3—source data 1*). Therefore, *Sulfitobacter* D7 grown in the pathogenicity-inducing media was indeed in a distinct transcriptional and metabolic state compared to the coexistence medium. Many transport genes were upregulated in Exp-CM +DMSP compared to Exp-CM: 99 genes, which constitute ~20% of *Sulfitobacter* D7 transport genes (*Figure 3*, *Figure 3—figure supplement 1*). An additional 39 genes were downregulated in response to DMSP addition to Exp-CM. Interestingly, also in MM +DMSP 42 transport genes were upregulated compared to MM, and 14 were downregulated. Namely, in both Exp-CM and MM, DMSP induced remodeling of the transporter repertoire. The fact that the addition of a single metabolite, that is DMSP, led to DE of a multitude of transporters for various metabolite classes demonstrates the signaling role of DMSP. When we examined the amount of DE transport genes that were shared between the comparisons of the DMSP-supplemented treatments, we found only eight genes that were DE in a similar manner (*Figure 3—figure supplement 1*). Namely, DMSP led to a shift in transporter gene expression in both media, but the identity of the DE transporters was unique for each medium. This strengthens our hypothesis that the DMSP signal affects bacterial gene expression, but the activation of the coexistence and pathogenicity transcriptional profiles depends on additional algal metabolites.

The differential effect of DMSP in Exp-CM and MM was especially notable in the expression of the DMSP transporters (betaine-carnitine-choline transporters; *Kiene et al., 1998*), which were significantly upregulated in Stat-CM and Exp-CM +DMSP, where DMSP was present at high concentrations compared to Exp-CM. However, the expression of these transporters was not affected by the addition of DMSP in the MM ±DMSP treatments. This was the same for the *DmdA* gene that encodes for the enzyme responsible for the first step of DMSP breakdown (*Curson et al., 2011*; *Figure 1—source data 2*). *DmdA* was barely expressed in MM +DMSP, albeit DMSP was present at high concentrations. Therefore, DMSP uptake and metabolism were prominent in *E. huxleyi*-derived CM, which contains additional algal metabolites that were not present in the MM. Taken together, DMSP has a strong signaling role and works in concert with additional algal metabolites to induce the coexistence- to pathogenicity-related gene expression in *Sulfitobacter* D7.

## Algal benzoate is a key metabolite for *E. huxleyi*-*Sulfitobacter* D7 coexistence

We searched in the *Sulfitobacter* D7 transcriptomic response to *E. huxleyi*-derived metabolites for evidence of involvement of additional algal factors, other than DMSP, in the regulation of the lifestyle switch from coexistence to pathogenicity. We revealed a plasmid-encoded degradation pathway of the aromatic compound benzoate that was highly expressed in CM samples (*Figure 4a–b*, *Figure*

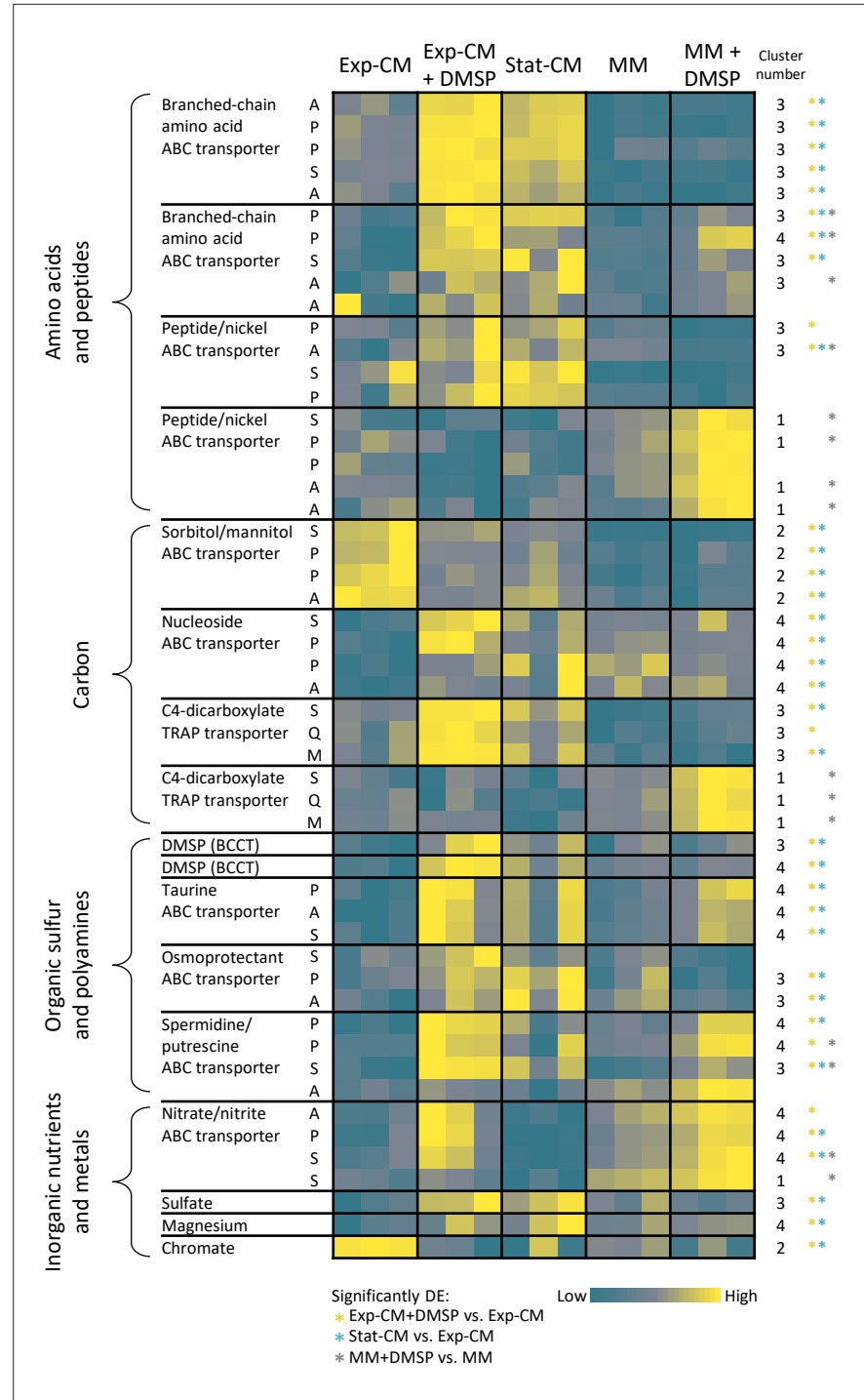

**Figure 3.** Remodeling of *Sulfitobacter* D7 transport systems in response to dimethylsulfoniopropionate (DMSP) and *Emiliania huxleyi*-derived metabolites. Heatmap of gene expression of representative transport genes for various metabolite classes. Each row represents one gene, and the blocks represent a complete transport system in which at least two genes were differentially expressed (DE) in the comparisons indicated on the bottom. The column 'cluster number' corresponds to the heatmap cluster in ***Figure 1d*** in which the gene is found. Colored * denotes in which comparison of the gene was significantly DE. Color intensity corresponds to the standardized expression across all samples (triplicates of each treatment are shown). Expression values are scaled by row. Expression and fold-change values are presented in ***Figure 3—source data 1***. ABC, ATP-binding cassette; TRAP, tripartite ATP-independent periplasmic; BCCT, betaine/carnitine/choline transporter. The letters correspond to

*Figure 3 continued on next page*

*Figure 3 continued*

the transport system components: A, ATP-binding; S, substrate-binding; P, permease; Q, dctQ subunit; M, dctM subunit.

The online version of this article includes the following source data and figure supplement(s) for figure 3:

**Source data 1.** *Sulfitobacter* D7 expression of DE transport genes.

**Figure supplement 1.** Differential expression of *Sulfitobacter* D7 transport genes in coexistence and pathogenic states and in response to dimethylsulfoniopropionate (DMSP).

---

*4—source data 1*). Aromatic compound degradation is a common metabolic feature in Roseobacters (*Buchan et al., 2004*; *Moran et al., 2007*). The metabolic intermediates of benzoate catabolism can be directed to β-ketoadipate, which is subsequently metabolized to form the tricarboxylic acid (TCA) cycle precursors, acetyl-CoA and succinate (*Harwood and Parales, 1996*). There are two pathways to metabolize benzoate to β-ketoadipate, through catechol (*Ben* genes) and through 4-hydroxybenzoate and protocatechuate (*BphA* and *PobA* genes) (*Figure 4—figure supplement 1*). An additional benzoate degradation pathway is through benzoyl-CoA (*Box* genes) (*Figure 4—figure supplement 1*; *Fuchs et al., 2011*). *Sulfitobacter* D7 harbors the pathway through catechol (*Figure 4a*). The *Ben* genes are localized in a gene cluster adjacent to a transcription factor *BenM*, which is known to regulate the expression of *BenABCD* and *CatAB* (*Figure 4b*; *Bundy et al., 2002*; *Collier et al., 1998*). All the genes in this benzoate-degradation gene cluster were expressed in CM treatments. Interestingly, the transporter of benzoate, which is encoded by a chromosomal gene, was also expressed in all CM, therefore it is not affected by the DMSP signal, as was observed for other transport systems (*Figure 3*). This suggests that *Sulfitobacter* D7 can assimilate and perceive algae-derived benzoate as a growth factor or signal regardless of the concentration of DMSP and may therefore be important in the initial coexistence phase.

To investigate if *Sulfitobacter* D7 can grow on benzoate, we inoculated the bacterium in MM supplemented with benzoate as a sole carbon source and found that it promoted bacterial growth in a dose-dependent manner (*Figure 4—figure supplement 2*). We further monitored the consumption of benzoate by growing the bacterium with 100 μM benzoate in MM. *Sulfitobacter* D7 grew by three orders of magnitude within 24 hr and consumed benzoate to an undetectable level, while the concentration of benzoate in the non-inoculated medium did not change significantly (*Figure 4c*). We further followed the concentration of benzoate in the medium of *E. huxleyi* cultures. We found that *E. huxleyi* produced and released benzoate to the medium when grown in mono-culture (*Figure 4d*). The concentration of benzoate in the algal culture's medium was in the tens of nM range. Nevertheless, the concentration of benzoate in the phycosphere is likely much higher (*Seymour et al., 2017*). Upon co-culturing with *Sulfitobacter* D7, algal benzoate was rapidly consumed by the bacteria. These results demonstrate that *Sulfitobacter* D7 can grow on alga-derived benzoate and benefit from this metabolite during interactions with *E. huxleyi*.

Bacterial degradation of various aromatic compounds is mostly directed to the β-ketoadipate pathway and eventually to the TCA cycle (*Harwood and Parales, 1996*). While this pathway seems to exist in many Roseobacters, the direct degradation of benzoate is limited to only few species (*Buchan et al., 2004*; *Newton et al., 2010*). We examined the prevalence of benzoate degradation and transport genes among phytoplankton-associated Proteobacteria. Specifically, we searched for genes encoding benzoate transporters and enzymes that directly metabolize benzoate through one of the three pathways (*Figure 4—figure supplement 1*, *Figure 5—source data 1*). We found that in addition to *Sulfitobacter* D7 another *Sulfitobacter* sp., CB2047, which was also isolated from an *E. huxleyi* bloom (*Ankrah et al., 2014*), may be able to utilize benzoate, namely, its genome encodes for both degradation and transport genes, while other *Sulfitobacter* genomes did not (*Figure 5a*). We found indications for benzoate utilization in the genomes of two additional Roseobacters, *Rhodobacteraceae* bacterium EhC02 and *Roseovarius indicus* EhC03, as well as *Sphingomonadales* bacterium EhC05, all isolated from *E. huxleyi* cultures (*Rosana et al., 2016*; *Figure 5a*). This was also evident in the genomes of several *Marinobacter* isolates, a genus known to be associated with *E. huxleyi* cultures (*Green et al., 2015*; *Orata et al., 2016*; *Rosana et al., 2016*). *Ruegeria pomeroyi* DSS-3 was the only bacterial strain included in our analysis, which is not directly associated to *E. huxleyi* but also holds the gene repertoire for benzoate utilization, consistent with previous observations (*Newton et al., 2010*).

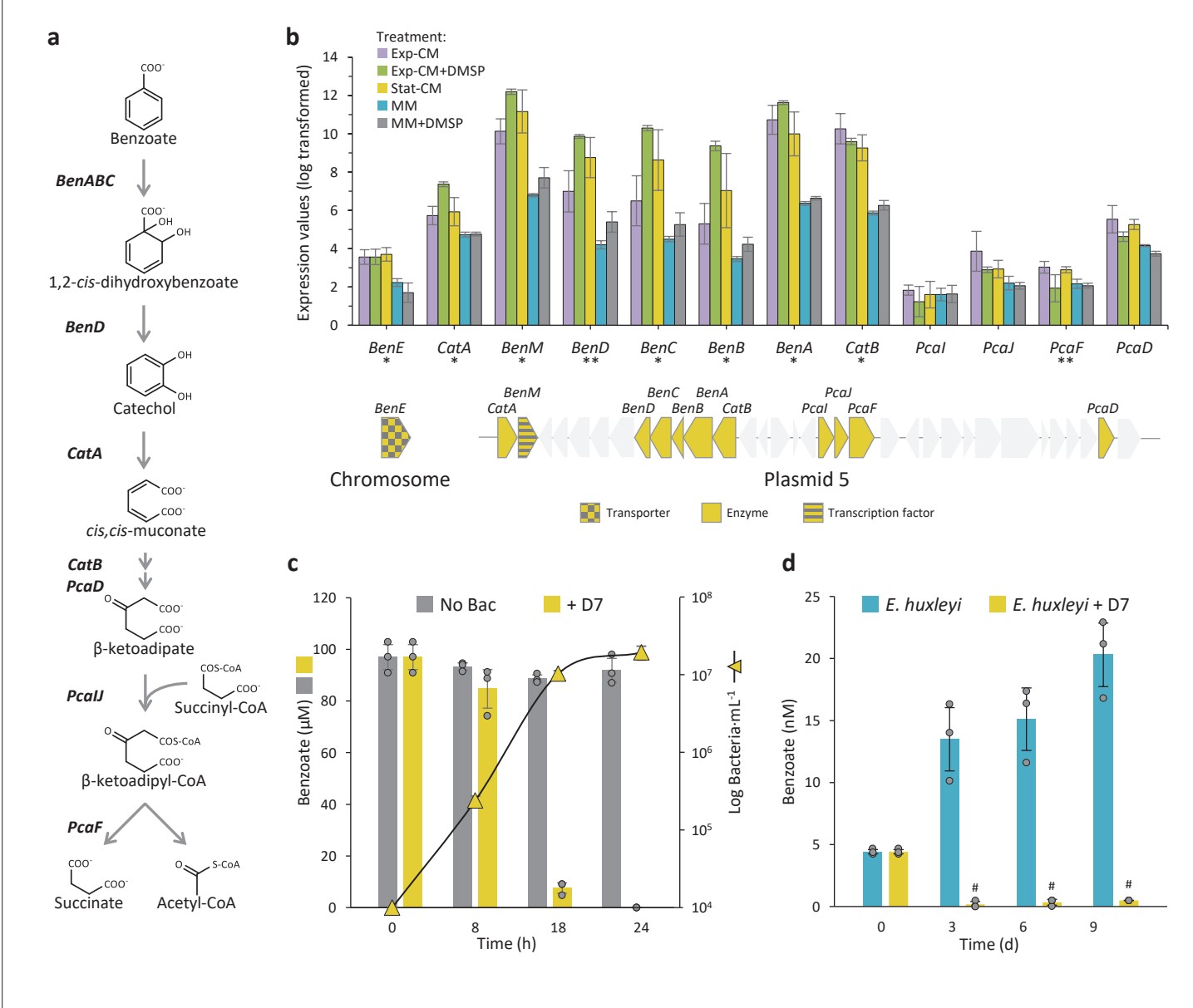

**Figure 4.** *Sulfitobacter* D7 encodes for a benzoate degradation pathway and uptakes *Emiliania huxleyi*-produced benzoate during interaction. (**a**) The benzoate degradation pathway in *Sulfitobacter* D7. The genes that encode for the enzymes mediating the subsequent transformations of benzoate to succinate and acetyl-CoA are denoted in bold. (**b**) Expression values of benzoate-related genes, which are localized on *Sulfitobacter* D7 plasmid 5 in a gene cluster, as indicated below the graph. The benzoate transporter, *BenE*, is encoded on the chromosome. Genes in gray are not related to benzoate. Results represent average ± SD (n=3). Expression and fold-change values are presented in *Figure 4—source data 1*. Genes marked with * were significantly differentially expressed (DE) between all conditioned media (CM) and minimal media (MM) treatments (Exp-CM vs MM, Exp-CM vs MM + DMSP, Exp-CM +DMSP vs MM, Exp-CM +DMSP vs MM +DMSP, Stat-CM vs MM, and Stat-CM vs MM +DMSP). Genes marked with ** were significantly DE between all CM and MM treatments but one comparison (for *BenD* Exp-CM vs MM +DMSP; for *PcaF* Exp-CM +DMSP vs MM). (**c**) Benzoate concentration (bars, left axis) and bacterial growth (triangles, right axis) in MM supplemented with 100 μM benzoate, as a sole carbon source, without inoculation (gray) and upon inoculation of *Sulfitobacter* D7 (yellow). Results represent average ± SD (n=3). p-Value <0.0001 for the difference in benzoate concentration between the 'No bac' and '+D7' treatments. No bacterial growth was observed in un-inoculated MM. (**d**) Extracellular benzoate concentration in the media of *E. huxleyi* CCMP379 cultures during growth in mono-culture (blue) or during co-culturing with *Sulfitobacter* D7 (yellow). # Benzoate was detected below the limit of quantification. Results represent average ± SD (n=3). p-Value <0.0001 for the difference in benzoate concentration between the '*E. huxleyi*' and '*E. huxleyi* +D7' treatments. Statistical differences in (c) and (d) were tested using a mixed effects model, with treatment and time as fixed effects, and replicate as a random effect. *BenABC*, benzoate 1,2-dioxygenase subunit alpha, beta, and reductase component, respectively; *BenD*, *cis*-1,2-dihydroxybenzoate dehydrogenase; *CatA*, catechol 1,2-dioxygenase; *CatB*, muconate cycloisomerase; *PcaD*,

*Figure 4 continued on next page*

*Figure 4 continued*

3-oxoadipate enol-lactonase; *PcaIJ*, 3-oxoadipate CoA-transferase, alpha and beta subunits, respectively; *PcaF*, 3-oxoadipyl-CoA thiolase, DMSP, dimethylsulfoniopropionate; CM, conditioned media; MM, minimal media; Exp-CM, exponential CM; Stat-CM, stationary CM.

The online version of this article includes the following source data and figure supplement(s) for figure 4:

**Source data 1.** *Sulfitobacter* D7 expression of benzoate transport and metabolism genes.

**Source data 2.** Benzoate concentration and bacterial growth in mono-cultures.

**Source data 3.** Extracellular benzoate concentration in the media of *Emiliania huxleyi* CCMP379 cultures.

**Figure supplement 1.** Bacterial benzoate degradation pathways.

**Figure supplement 2.** Benzoate promotes *Sulfitobacter* D7 growth.

**Figure supplement 3.** Detection and quantification of benzoate by two Liquid chromatography–mass spectrometry (LC-MS) methods.

This suggests that benzoate produced by *E. huxleyi* can mediate interactions with several bacteria that consume and benefit from this metabolite.

Interestingly, there were several *Sulfitobacter* strains whose genome did not indicate benzoate utilization (*Figure 5a*). We assessed *S. pontiacus* DSM 10014 and *S. brevis* DSM 11443 for growth on benzoate as a sole carbon source in MM. As anticipated from the genome analysis, both strains did not grow on benzoate, while *Sulfitobacter* D7 did (*Figure 5b*). We also evaluated growth on succinate as sole carbon source and all three strains grew on it. Intriguingly, *Sulfitobacter* D7 grew significantly better on benzoate than on succinate, demonstrating that benzoate is more preferable as a carbon source.

To assess whether the ability to utilize benzoate by bacteria affects their interaction with *E. huxleyi*, we co-cultured the three *Sulfitobacter* strains with the alga. As expected, *Sulfitobacter* D7 showed the lifestyle switch from coexistence to pathogenicity (*Figure 5c*). *S. pontiacus* DSM 10014 and *S. brevis* DSM 11443 remained in coexistance with *E. huxleyi* and did not display a lifestyle switch (*Figure 5c*). All three *Sulfitobacter* strains grew during the interaction with the alga, but *Sulfitobacter* D7 grew significantly better than the other strains, already in the coexistence phase (*Figure 5c*). This suggests that the metabolic exchange of benzoate between *E. huxleyi* and *Sulfitobacter* D7 may be important for the bacterial lifestyle switch.

To examine the role of benzoate in the DMSP-induced lifestyle switch of *Sulfitobacter* D7, we followed the dynamics of co-cultures to which we externally added benzoate and DMSP. As expected, addition of DMSP induced the bacterial lifestyle switch to pathogenicity resulting in a faster algal decline already at day 4, as compared to day 10 when no DMSP was added (*Figure 6*). Addition of only benzoate did not alter the dynamics of algal growth, but bacterial growth was enhanced at day 4, compared to the non-supplemented and DMSP-supplemented co-cultures (*Figure 6c*). This goes along with the observation of benzoate being a good carbon source for *Sulfitobacter* D7 growth (*Figure 4c*, *Figure 5b*). When we applied both benzoate and DMSP in equimolar concentrations, the algal cultures declined from day 8 onward, that is, the induction of pathogenicity by DMSP was delayed by 4 days, as compared to the DMSP addition in the absence of benzoate. This suggests that benzoate hinders the pathogenicity-inducing effect of DMSP. To further support this observation, we examined the interaction of *Sulfitobacter* D7 with a different *E. huxleyi* strain, CCMP2090, with which the bacterial lifestyle switch does not naturally occur (*Figure 6—figure supplement 1*). Only when we added external DMSP it led to the decline of *E. huxleyi* abundance (*Figure 6—figure supplement 1*), consistent with our previous findings (*Barak-Gavish et al., 2018*). Upon addition of both benzoate and DMSP, algal growth was not compromised, although *Sulfitobacter* D7 and DMSP were present at high concentrations (*Figure 6—figure supplement 1*). Intriguingly, bacterial abundance in the benzoate-added treatments was higher than in the pathogenicity-inducing DMSP treatment (*Figure 6c*, *Figure 6—figure supplement 1*), which demonstrates that the onset of pathogenicity is decoupled from bacterial density. Taken together, these results suggest that benzoate is important to maintain *E. huxleyi*-*Sulfitobacter* D7 coexistence and can hinder the onset of bacterial pathogenicity. This observation strengthens our conclusion from the transcriptomics analysis that DMSP signaling in *Sulfitobacter* D7 depends on additional alga-derived metabolites which affect the bacterial lifestyle switch during interaction with *E. huxleyi*.

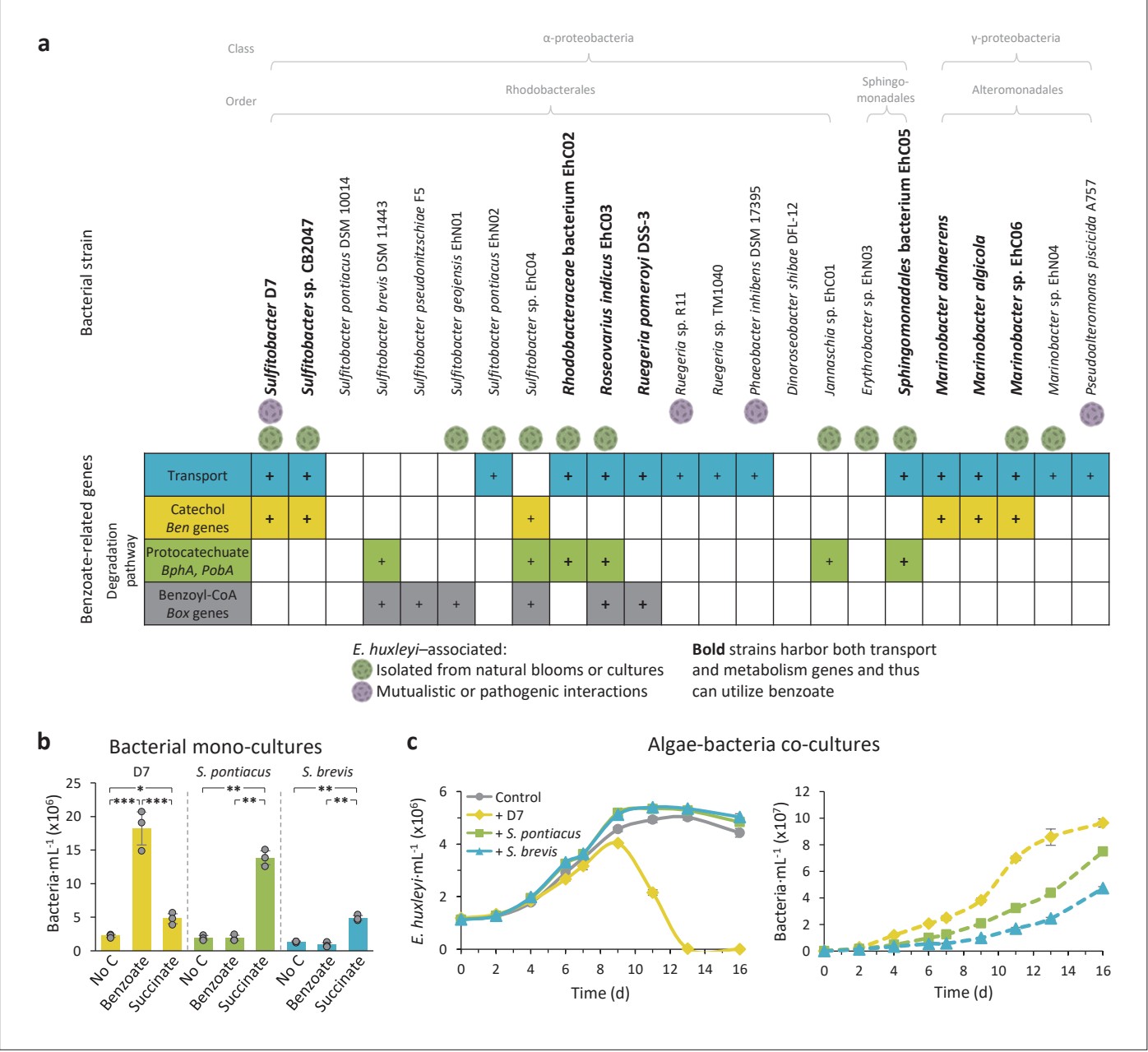

**Figure 5.** Benzoate utilization may impact the bacterial lifestyle switch. (**a**) Presence of benzoate transport and degradation genes in genomes of phytoplankton-associated Proteobacteria. *Emiliania huxleyi*-associated bacteria are denoted. Colored tiles depict the presence of the genes. Bacterial strains highlighted in bold possess genes for both transport and degradation of benzoate. Bacterial benzoate degradation pathways are elaborated in *Figure 4—figure supplement 1*. The full data of the presence of all query genes in the genomes of the bacteria is presented in *Figure 5—source data 1*. The query genes are listed in Key resources table. (**b**) Bacterial growth after 24 hr in minimal media with no carbon source or supplemented with 100 µM benzoate or succinate, as a sole carbon source, of *Sulfitobacter* D7 (yellow) and two additional *Sulfitobacter* strains (*Sulfitobacter pontiacus* DSM 10014 [green] and *Sulfitobacter brevis* DSM 11443 [blue]) that do not possess the genes required for benzoate utilization. Results represent average ± SD (n=3). Statistical differences were tested using two-way analysis of variance, followed by Tukey's post-hoc test. * p-Value <0.05, ** p-value <0.01 *** p-value <0.0001. (**c**) Time course of *E. huxleyi* CCMP379 and bacterial abundance (left and right panels, respectively) in algal mono-cultures (gray) or during co-culturing with *Sulfitobacter* D7 (yellow), *S. pontiacus* (green), or *S. brevis* (blue). Results represent average ± SD (n=3). Statistical differences were tested using repeated-measures analysis of variance, followed by a Tukey post hoc test. p-Value <0.0001 for the difference in *E. huxleyi* and bacterial growth between all treatments, except for *E. huxleyi* growth in the treatments '+*S. pontiacus*' compared to '+*S. brevis*', which was not significant.

The online version of this article includes the following source data for figure 5:

*Figure 5 continued on next page*

*Figure 5 continued*

**Source data 1.** Presence of benzoate degradation and transport genes in genomes of phytoplankton-associated proteobacteria.

**Source data 2.** Bacterial growth in MM with different carbon sources.

**Source data 3.** *Emiliania huxleyi* CCMP379 and bacterial abundances.

## Discussion

### Signaling role of DMSP and other algal metabolites in the lifestyle switch of *Sulfitobacter* D7

In this study, we aimed to unravel the molecular basis for the lifestyle switch from coexistence to pathogenicity in *Sulfitobacter* D7 during interaction with the bloom-forming alga *E. huxleyi*. We substantiated the signaling role of algal DMSP that mediates the shift toward pathogenicity by mapping the transcriptional profiles of *Sulfitobacter* D7 in response to DMSP and other algal metabolites. However, DMSP signaling in medium that lacked *E. huxleyi*-derived metabolites (i.e. MM +DMSP) had a different effect on *Sulfitobacter* D7 transcriptome. We propose that the signaling role of DMSP that mediates the coexistence to pathogenicity lifestyle switch in *Sulfitobacter* D7 depends on other infochemicals produced by *E. huxleyi*. DMSP is a ubiquitous infochemical produced by many phytoplankton species

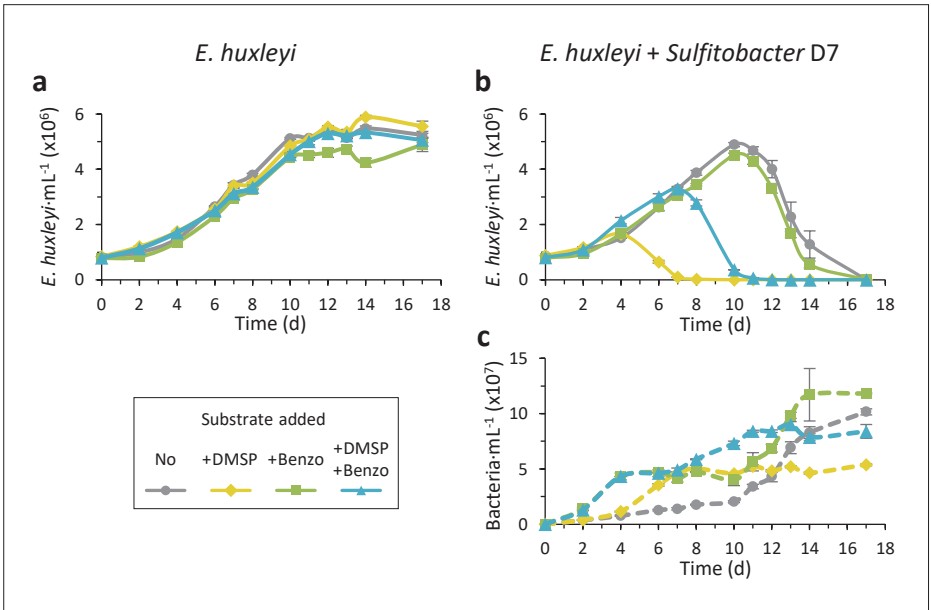

**Figure 6.** Benzoate hinders the dimethylsulfoniopropionate (DMSP)-induced pathogenicity of Sulfitobacter D7 against *Emiliania huxleyi*. Time course of *E. huxleyi* CCMP379 and bacterial abundance (smooth and dashed lines, respectively) in algal mono-cultures (**a**) or during co-culturing with *Sulfitobacter* D7 (**b-c**). No bacterial growth was observed in algal mono-cultures. Cultures were supplemented at day 0 with 100 μM of DMSP (yellow), benzoate (benzo, green), DMSP and benzoate (blue), or none (gray). The presence of benzoate hindered the pathogenicity-inducing effect of DMSP. Results represent average ± SD (n=3). Statistical differences were tested using repeated-measures analysis of variance, followed by a Tukey post hoc test. p-Value <0.0001 for the difference in *E. huxleyi* growth in the treatment of '+D7+DMSP' and 'D7 +DMSP + Benzoate' compared to all other treatments. p-Value <0.0001 for the differences in bacterial growth in the treatments '+D7+benzoate' and '+D7+Benzoate + DMSP' compared to only '+D7'.

The online version of this article includes the following source data and figure supplement(s) for figure 6:

**Source data 1.** Time course of *Emiliania huxleyi* CCMP379 abundance in mono-cultures supplemented with different substrates.

**Source data 2.** Time course of *Emiliania huxleyi* CCMP379 and *Sulfitobacter* D7 abundance in co-cultures supplemented with different substrates.

**Figure supplement 1.** Benzoate is a key metabolite for maintaining *Emiliania huxleyi-Sulfitobacter* D7 coexistence.

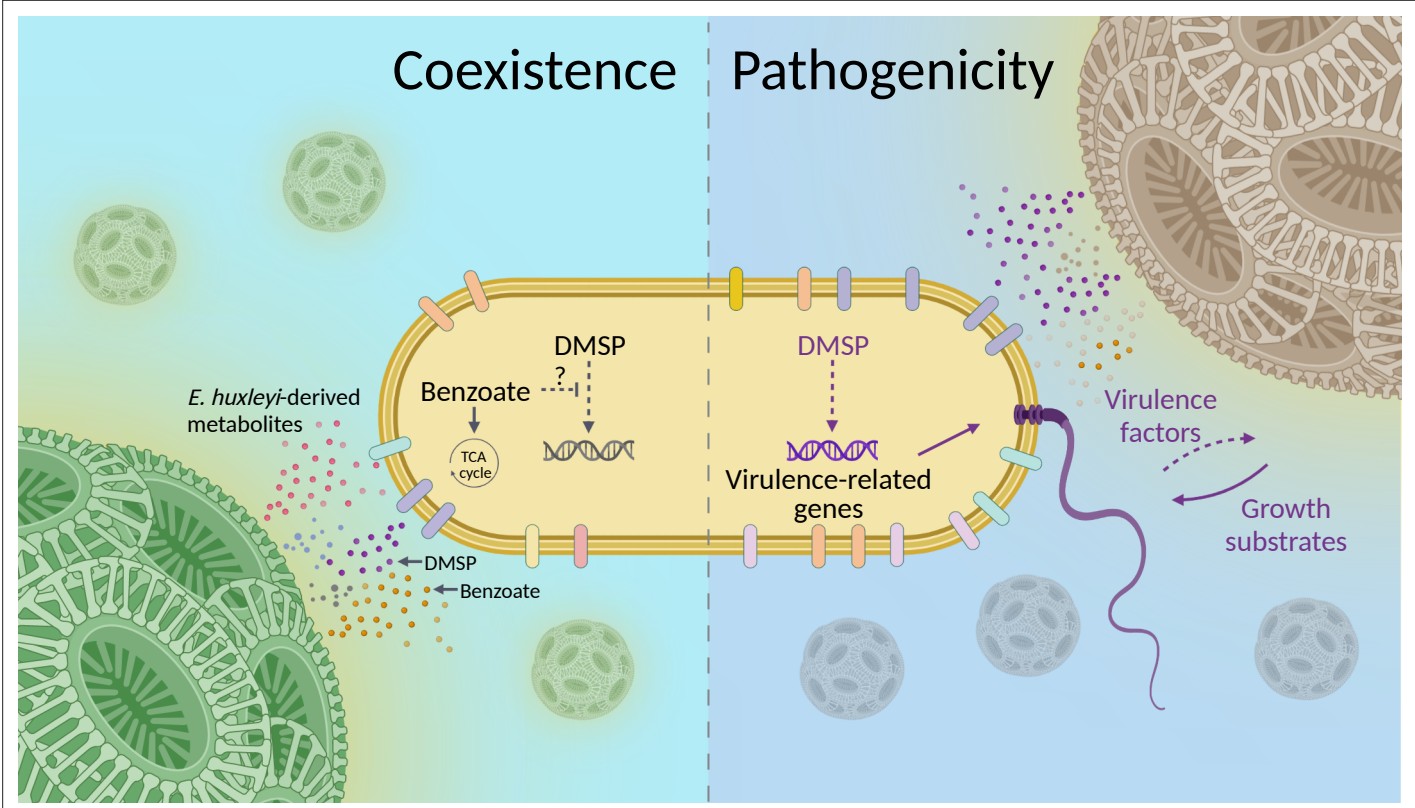

**Figure 7.** Conceptual model of the lifestyle switch of *Sulfitobacter* D7 in response to *Emiliania huxleyi*-derived metabolites. During its interactions with *E. huxleyi*, *Sulfitobacter* D7 exhibits a lifestyle switch from coexistence to pathogenicity. In the coexistence phase, *E. huxleyi* secretes to the phycosphere various metabolites such as benzoate, dimethylsulfoniopropionate (DMSP), and other growth substrates, which bacteria can uptake and consume for growth. Based on the observation that benzoate hindered the pathogenicity-inducing effect of DMSP, we hypothesize that such energy-rich metabolic currencies hinder DMSP signaling in *Sulfitobacter* D7. When the algal physiological state is compromised, for example, stationary phase, the amount of available growth substrates decreases, due to bacterial consumption and less secretion by the alga. In this context, high concentration of algal DMSP acts as a signal that alters the transcriptional profiles of the bacterium and leads to high expression of pathogenicity-related genes, such as flagellar and transport genes, and yet unknown virulence factors that kill *E. huxleyi* cells. This leads to a surge of alga-derived growth substrates that are taken up efficiently by *Sulfitobacter* D7. The flagellum can mediate the dispersal of *Sulfitobacter* D7 and to forage for an alternative host.

as well as some bacteria (**Curson et al., 2017**), making it a prevalent signaling molecule that mediates microbial interactions in the marine environment. Therefore, it is likely that other algal metabolites are involved in the recognition of the specific phytoplankter host by bacteria, thus ensuring specificity in DMSP signaling during alga-bacteria interactions. In natural environments, where many microbial species are present simultaneously, such a mechanism can ensure that bacteria will invest in altering gene expression and metabolic remodeling only when the right algal partners are present.

We revealed that the alga-derived aromatic compound benzoate plays a pivotal role in *Sulfitobacter* D7-*E. huxleyi* interaction by maintaining the coexistence, even when DMSP is present at high concentrations (**Figure 6**, **Figure 6—figure supplement 1**). Benzoate also acts as an efficient bacterial growth factor serving as a carbon source (**Figure 4**). These observations provide a possible explanation for the switch in bacterial behavior from coexistence to pathogenicity. During the interaction, *E. huxleyi* provides benzoate and other growth substrates to *Sulfitobacter* D7, which uptakes and consumes them (**Figure 7**). We propose that as long as *Sulfitobacter* D7 benefits from the interaction with *E. huxleyi* by receiving beneficial growth substrates, it will maintain a coexisting lifestyle. When less growth substrates are provided by the alga, the opportunistic pathogen will switch to killing the algal host, which will in turn lead to a surge of intracellular *E. huxleyi*-derived metabolites that *Sulfitobacter* D7 can benefit from (**Figure 7**). Studies on phytoplankton exudation of organic matter demonstrated that algae release more organic matter in stationary phase, but the chemical composition is different from that of exponential growth (**Barofsky et al., 2009**; **Jensen, 1984**). In nutrient limiting

conditions, which often occur in stationary phase, the organic matter exuded by phytoplankton is less favorable for bacterial uptake and consumption for growth (*Obernosterer and Herndl, 1995*). In such a chemical context, high concentrations of algae-derived infochemicals, for example, DMSP (*Barak-Gavish et al., 2018*) or *p*-coumaric acid (*Seyedsayamdost et al., 2011*), can be perceived by bacteria and signal that the physiological state of the algal host is deteriorating. Namely, by sensing the changes in the metabolic composition of the phycosphere during the interaction, *Sulfitobacter* D7 executes its pathogenicity against a compromised *E. huxleyi* population. Therefore, the initial metabolic exchange in the coexistence phase is a prerequisite for the onset of bacterial pathogenicity.

The ability to utilize benzoate is shared among bacterial strains that are associated with *E. huxleyi* in the natural environment and in cultures (*Figure 5a*; *Green et al., 2015*; *Orata et al., 2016*; *Rosana et al., 2016*; *Vincent et al., 2021*). Since benzoate can act as an antibacterial compound (*Amin and Abolmaaty, 2020*; *Haque et al., 2005*), we propose that secretion of benzoate by *E. huxleyi* can select for bacteria that specialize on this compound and is therefore important for the establishment of a coexistence phase. Similarly, the diatom *Asterionellopsis glacialis* produces two secondary metabolites that select for specific bacteria and affect their behavioral response (*Shibl et al., 2020*). Bacterial sensing of general phytoplankton-derived compounds (e.g. DMSP) together with additional more selective compounds (e.g. benzoate) can ensure the recognition of the algal host by the bacteria within the phycosphere. This can increase the specificity of an interaction and ensure fine-tuning of the behavior of microorganisms by regulating gene expression. This is especially relevant for DMSP that has diverse functions in bacteria (*Barak-Gavish et al., 2018*; *Kessler et al., 2018*; *Miller et al., 2004*; *Miller and Belas, 2004*; *Seymour et al., 2010*). Molecular mechanisms in bacteria that integrate information perceived by various chemical signals include catabolite repression and two-component systems, which can also play a role in regulating bacterial pathogenicity (*Beier and Gross, 2006*; *Görke and Stülke, 2008*).

## The lifestyle switch of *Sulfitobacter* D7 from coexistence to pathogenicity

Our experimental setup demonstrated that *Sulfitobacter* D7 grown in pathogenicity-inducing media are in a different transcriptional state than in coexistence medium, which corresponds to the behavioral switch during co-culturing with *E. huxleyi* (*Figure 1*). Many transport systems were DE, mainly upregulated, when *Sulfitobacter* D7 was in pathogenic state compared to the coexistence state (*Figure 3*). Since bacteria often exert their pathogenicity as a means to access nutrients released from the host, it is likely that in this mode *Sulfitobacter* D7 will maximize uptake and assimilation of metabolites released by dying *E. huxleyi* cells. High expression of transporters for branched-chain amino acids, C4 carbohydrates, DMSP, taurine, and polyamines can facilitate the efficient uptake of these energy-rich metabolites (*Figure 3*, *Figure 3—source data 1*). Upregulation of transport genes for these metabolic currencies in response to DMSP was also demonstrated in *R. pomeroyi* DSS-3, a Roseobacter often used to study metabolic exchange between bacteria and phytoplankton (*Bürgmann et al., 2007*; *Durham et al., 2015*; *Landa et al., 2017*).

During the pathogenic lifestyle there was upregulation of flagellar genes, which was functionally validated by motility assays (*Figure 2*). While DMSP is a known chemoattractant and therefore mediates the establishment of bacterial interactions with algae (*Miller et al., 2004*; *Seymour et al., 2010*), we speculate that this is not the case for *Sulfitobacter* D7, since its genome does not encode for known chemotaxis genes. We propose that the increased motility in response to DMSP in the pathogenic mode can serve as an ecological strategy to escape from competition with other bacteria in the phycosphere (*Yawata et al., 2014*). *E. huxleyi* cell death, induced by *Sulfitobacter* D7, likely leads to a surge of intracellular metabolites that may attract other bacteria. The upregulation of transport systems together with flagellar motility can enable efficient substrate uptake by *Sulfitobacter* D7 and swimming away to forage for alternative metabolically active hosts. Such an 'eat-and-run' strategy can be ecologically beneficial by facilitating the evasion from competition.

Upregulation of flagellar genes was also demonstrated during the mutualistic to pathogenic lifestyle switch of the Roseobacter *Dinoroseobacter shibae* during interaction with a dinoflagellate algal host (*Wang et al., 2015*). Even though *Sulfitobacter* D7 motility was increased in the pathogenic mode (*Figure 2*), the involvement of the flagellum may be by other functions that mediate bacterial virulence (*Chaban et al., 2015*); that is, flagella can mediate biofilm formation and attachment to

surfaces (*Li et al., 2016*). Additionally, the flagellar type 3 secretion system, which is found in the basal body and necessary for secretion of the components needed for flagellum assembly, can also be used as an export system for effector proteins in pathogenic bacteria (*Diepold and Armitage, 2015*). In this manner, pathogenic bacteria may utilize the flagellum for multiple functions important for pathogenicity against their hosts and subsequent dispersal.

The mechanism of *Sulfitobacter* D7 pathogenicity against *E. huxleyi* remains to be discovered. The genome of *Sulfitobacter* D7 encodes numerous cellular machineries, such as type 2 secretion system (T2SS), Flp (fimbrial low-molecular-weight protein) pilus, and two type 4 secretion systems, which can potentially mediate cell-cell interactions and possibly bacterial virulence (*Backert and Meyer, 2006*; *Cianciotto and White, 2017*; *Ku et al., 2018*; *Tomich et al., 2007*). While the Flp pilus and T4SS are common features encoded in Roseobacter genomes, T2SS is less prevalent (*Frank et al., 2015*; *Slightom and Buchan, 2009*). This may hint for a unique mode of pathogenicity in *Sulfitobacter* D7 and requires further investigation.

## Ecological context of bacterial lifestyle switches during algal blooms

Bacterial lifestyle switches are evident in several model systems of phytoplankton-bacteria interactions; however, the ecological significance of such modes of interaction in the natural environment is elusive. In this study, we provide a contextual framework for the switch from coexistence to pathogenicity – metabolite depletion in the phycosphere. During a phytoplankton bloom, heterotrophic bacteria can support the growth of the algae and benefit from organic matter released to the phycosphere. As the bloom progresses, various factors, such as nutrient depletion, viral infection, and grazing, can compromise the algal population and its ability to provide essential metabolic currencies for optimal bacterial growth. We propose that bacteria can sense the host's physiological state, by infochemicals secreted from stressed algae, and switch their behavior to pathogenicity. This will result in algal cell death and bacterial proliferation, which could eventually contribute to the bloom demise. Therefore, phytoplankton-associated opportunistic bacterial pathogens constitute an underappreciated component in the regulation of algal blooms dynamics. Investigating the dynamic microscale interactions of such bacteria with phytoplankton and the metabolic crosstalk that mediates them can provide insights into their impact on large scale biogeochemical processes in the marine environment.

## Materials and methods

**Key resources table**

| Reagent type (species) or resource | Designation | Source or reference | Identifiers | Additional information |
|---|---|---|---|---|
| Gene (*Acinetobacter* sp. strain ADP1) | $BenK_1$, Benzoate transporter | *Collier et al., 1997* | AAC46425.1 | |
| Gene (*Pseudomonas putida* KT2440) | $BenK_2$, Benzoate transporter | *Nishikawa et al., 2008* | AAN68773.1 | |
| Gene (*Pseudomonas putida* KT2440) | $BenE_1$, Benzoate transporter | *Nishikawa et al., 2008* | AAN67649.1 | |
| Gene (*Pseudomonas putida* KT2440) | $BenE_2$, Benzoate transporter | *Nishikawa et al., 2008* | AAN68775.1 | |
| Gene (*Pseudomonas putida* KT2440) | *BenF*, Benzoate transporter | *Nishikawa et al., 2008* | AAN68776.1 | |
| Gene (*Acinetobacter* sp. strain ADP1) | *BenA*, Benzoate 1,2-dioxygenase subunit alpha | *Collier et al., 1998* | AAC46436.2 | Benzoate degradation pathway: Catechol |
| Gene (*Acinetobacter* sp. strain ADP1) | *BenB*, Benzoate 1,2-dioxygenase subunit beta | *Collier et al., 1998* | AAC46437.1 | Benzoate degradation pathway: Catechol |

*Continued on next page*

*Continued*

| Reagent type (species) or resource | Designation | Source or reference | Identifiers | Additional information |
|---|---|---|---|---|
| Gene (*Acinetobacter* sp. strain ADP1) | *BenC*, Benzoate 1,2-dioxygenase electron transfer component | *Collier et al., 1998* | AAC46438.1 | Benzoate degradation pathway: Catechol |
| Gene (*Aspergillus niger*) | *BphA*, Benzoate 4-monooxygenase | *van Gorcom et al., 1990* | P17549.1 | Benzoate degradation pathway: Protocatechuate |
| Gene (*Acinetobacter* sp. strain ADP1) | *PobA*, 4-hydroxybenzoate 3-monooxygenase | *Brzostowicz et al., 2003* | WP_004926674.1 | Benzoate degradation pathway: Protocatechuate |
| Gene (*Azoarcus evansii*) | Benzoate CoA-ligase | *Mohamed et al., 2001; Gescher et al., 2002* | AAN39371.1 | Benzoate degradation pathway: Benzoyl-CoA |
| Gene (*Azoarcus evansii*) | *BoxA*, Benzoyl-CoA oxygenase component A | *Gescher et al., 2002* | AAN39377.1 | Benzoate degradation pathway: Benzoyl-CoA |
| Gene (*Azoarcus evansii*) | *BoxB*, Benzoyl-CoA oxygenase component B | *Gescher et al., 2002* | AAN39376.1 | Benzoate degradation pathway: Benzoyl-CoA |
| Strain, strain background (*Emiliania huxleyi*) | CCMP379 | National Center for Marine Algae (NCMA) | | |
| Strain, strain background (*Emiliania huxleyi*) | CCMP2090 | National Center for Marine Algae (NCMA) | | |
| Strain, strain background (*Sulfitobacter* D7) | | *Barak-Gavish et al., 2018* | | |
| Strain, strain background (*Sulfitobacter pontiacus*) | DSM 10014 | The Leibniz Institute DSMZ | | |
| Strain, strain background (*Sulfitobacter brevis*) | DSM 11443 | The Leibniz Institute DSMZ | | |
| Chemical compound, drug | (2-Carboxyethyl) dimethylsulfonium chloride 97% (DMSP) | Holland Moran | HC003BMH | |
| Chemical compound, drug | Sodium benzoate | Sigma-Aldrich | B3420 | |
| Software, algorithm | R | *R Development Core Team, 2021* | | |
| Software, algorithm | *Partek Genomics Suite* software, v7.0 | *Partek Inc, 2020* | | |
| Software, algorithm | Heatmapper | *Babicki et al., 2016* | | |
| Software, algorithm | g:Profiler | *Raudvere et al., 2019* | | |

## *E. huxleyi* cultures maintenance and co-culturing with bacteria

*E. huxleyi* strains were purchased from the National Center for Marine Algae and maintained in filtered sea water (FSW). CCMP379 was cultured in f/2 medium (-Si) (*Guillard and Ryther, 1962*), and CCMP2090 was cultured in k/2 medium (-Tris, -Si) (*Keller et al., 1988*). Cultures were incubated at 18°C with a 16:8 hr, light:dark illumination cycle. A light intensity of 100 µmol photons $m^{-2}$ $s^{-1}$ was provided by cool white LED lights. Algal cultures were kept in antibiotics to keep them axenic. Prior to inoculation of bacteria, the algal cultures were transferred at least three times to ensure that antibiotics were removed. We routinely checked for bacterial contamination in the algal cultures by flow cytometry. For all co-culturing experiments *E. huxleyi* cultures were inoculated at early exponential growth phase (4–8·$10^5$ cell $mL^{-1}$) with $10^3$ bacteria $mL^{-1}$ at t=0 days. When noted, DMSP or benzoate was added at t=0 days at final concentration of 100 µM.

## Enumeration of algae and bacteria abundances by flow cytometry

Flow cytometry analyses were performed on Eclipse iCyt flow cytometer (Sony Biotechnology Inc, Champaign, IL, USA) equipped with 405 and 488 nm solid-state air-cooled lasers, and with standard

optic filter set-up. *E. huxleyi* cells were identified by plotting the chlorophyll fluorescence (663–737 nm) against side scatter and were quantified by counting the high-chlorophyll events. For bacterial counts, samples were fixed with a final concentration of 0.5% glutaraldehyde for at least 30 min at 4°C, then plunged into liquid nitrogen and stored at −80°C until analysis. After thawing, samples were stained with SYBR gold (Invitrogen) that was diluted 1:10,000 in Tris–EDTA buffer, incubated for 20 min at 80°C, and cooled to room temperature. Samples were analyzed by flow cytometry (ex: 488 nm; em: 500–550 nm).

## Bacterial growth media

The CM, Exp-CM and Stat-CM, were obtained from exponential and stationary *E. huxleyi* CCMP379 mono-cultures (*Table 1*), respectively, by gentle gravity filtration on Whatman glass microfiber GF/C filters (pore size of 1.2 µm). This method was chosen to prevent lysis of algal cells during the procedure and thus ensuring that only extracellular algae-derived metabolites, infochemicals, and other components will reside in the media. CM were subsequently filtered through 0.22 µm using Stericup vacuum filters. Exp-CM and Stat-CM were harvested on the same day of the experiment. When indicated, 100 µM DMSP was added to Exp-CM, herein Exp-CM +DMSP. This concentration mimics that present in the phycosphere of *E. huxleyi* in stationary phase (*Barak-Gavish et al., 2018*; *Seymour et al., 2017*). MM was based on artificial sea water (ASW) (*Goyet and Poisson, 1989*) supplemented with basal medium (-Tris) (containing essential nutrients) (*Baumann and Baumann, 1981*), vitamin mix (*González et al., 1997*), 0.5 mM NaNO3, and metal mix of k/2 medium (*Keller et al., 1988*). For the transcriptome experiment, MM were supplemented with 1 gr L$^{-1}$ glycerol. When indicated, 100 µM DMSP was added to MM, herein MM +DMSP.

## Bacterial inoculation into growth media and *E. huxleyi* cultures

Bacteria were inoculated from a glycerol stock (kept at −80°C) into marine broth (MB) (Difco 2216) or ½YTSS (2 g of yeast extract, 1.25 g of tryptone, and 20 g of sea salts [Sigma-Aldrich] dissolved in 1 liter of double distilled water) and grown over-night at 28°C, 160 rpm. Bacteria were washed three times in FSW or ASW by centrifugation (10,000 g, 1 min). Bacterial inocula were counted by flow cytometry and 10$^4$ bacteria mL$^{-1}$ were inoculated to CM or MM, and 10$^3$ bacteria mL$^{-1}$ were inoculated to *E. huxleyi* cultures.

## *Sulfitobacter* D7 transcriptome

### Library preparation and sequencing

Experimental setup is elaborated in *Figure 1b*. Samples for bacterial growth and RNA were taken at t=24 hr. Bacterial cell pellets were obtained from 120 mL (MM treatments) or 160 mL (CM treatments) cultures by two-step centrifugation: 10,000 g, 10 min followed by 14,000 g, 10 min, all at 4°C. Pellets were flash frozen in liquid nitrogen and stored at −80°C until further analysis. RNA extraction was carried out using the RNeasy Plant Mini Kit (Qiagen, Hilden, Germany). For disruption of cell pellets we used the OmniLyse lysis kit (Claremontbio). The rest of the RNA extraction protocol was according to manufacturer's instruction. Library preparation was carried out according to the RNA-seq protocol developed by *Avraham et al., 2016*. Briefly, DNA was removed using TURBO DNase (Ambion), RNA was fragmented and phosphorylated (at the 3' prime) using FastAP thermosensitive alkaline phosphatase (Thermo Scientific). RNA from each sample was ligated with unique RNA barcoded adaptors at the 3', ensuring the strandedness of each transcript, using T4 RNA ligase 1 (NEB). RNA samples were pooled and treated with RiboZero (Gram-Negative Bacteria) kit (Illumina) following manufacturer's instructions in order to remove ribosomal RNA. Samples were reverse transcribed using Affinity-Script RT Enzyme (Agilent) to form cDNA and amplified by PCR. The libraries were sequenced at the Weizmann Institute of Science Core Facilities on an Illumina NextSeq500 high output v2 kit (paired end, 150 cycles).

### Transcriptome analysis

Raw reads (64.5 million) were quality trimmed using Cutadapt (*Martin, 2011*) (-q 20 m 20) in addition to removal of adapters. Reads were mapped to *Sulfitobacter* D7 genome assembly (GCA_003611275.1) using Bowtie2 (*Langmead and Salzberg, 2012*) in end-to-end mode, and reads were counted on genes using HTseq, in the strict mode (*Anders et al., 2015*). Final reads per sample can be found

in *Figure 1—source data 4*. Gene expression was quantified using DESeq2 (*Love et al., 2014*; *Figure 1—source data 2*). DE genes were selected as genes with adjusted p-value <0.05, and |fold change|>2, and basemean >10 (the average of the normalized count values, dividing by size factors, taken over all samples). Priciple component analysis and similarity between samples were calculated using DESeq2 and visualized using RStudio 3.5.0. Heatmaps of gene expression were calculated using the log-normalized expression values (rld), with row standardization (scaling the means of a row to zero, with standard deviation of 1), and visualized using *Partek Genomics Suite* software, v7.0 (*Partek Inc, 2020*), Heatmapper (*Babicki et al., 2016*), and Excel. The data has been deposited in NCBI's Gene Expression Omnibus (GEO) and is available through GEO series accession number GSE193203.

## Functional enrichment in KEGG pathways
DE genes in the comparisons Exp-CM +DMSP vs. Exp-CM, Stat-CM vs. Exp-CM, and MM +DMSP vs. MM were clustered using k-means analysis. For each cluster, enriched KEGG pathways (with Padj <0.01) were calculated by g:Profiler (*Raudvere et al., 2019*), using a customized reference which was constructed from *Sulfitobacter* D7 specific KEGG pathways.

## *Sulfitobacter* D7 genome mining and manual annotation
The automatic NCBI Prokaryotic Genome Annotation Pipeline was used for *Sulfitobacter* D7 genome functions prediction (*Ku et al., 2018*). We manually validated the function of genes related to DMSP metabolism, transport, benzoate degradation, and flagella assembly by cross examining their annotation using KEGG, COG, and IMG/M. For genes with no or inconsistent annotation, we also searched for functional domains using the Conserved Domain Database (CDD), and we ran BLAST using genes with known functions to validate the annotation.

### Transport genes
The automatic annotations of transport genes were manually validated by ensuring that the genes were annotated as such by at least two automatic annotation platforms and by CDD search. Since transport systems are organized in operon-like structures, we examined the genes adjacent to the transport genes and manually annotated these additional transport genes. The transporters presented in the heatmap (*Figure 3*) are the full transport systems that at least two of the genes in each system were DE. The Venn diagrams (*Figure 3—figure supplement 1*) contain only the transport genes that were significantly DE. The substrates for each transport system were inferred automatically, therefore, the exact substrates were not experimentally validated.

### Benzoate degradation genes
*Sulfitobacter* D7 benzoate degradation pathway was reconstructed using the KEGG mapping tool (*Kanehisa et al., 2022*). All the genes in the catechol branch of benzoate degradation (*Figure 4—figure supplement 1*) were found, except Muconolactone isomerase *CatC*. For the visualization of the organization of the genetic locus of benzoate-related genes we utilized the IMG/M platform (*Chen et al., 2017*).

### Flagellar genes
We manually validated the annotation of all the flagella genes in *Sulfitobacter* D7 genome and found most of the genes, except for three: *FliQ*, *FliJ*, and *FliD* (*Chevance and Hughes, 2008*). For the visualization of the organization of the genetic locus of flagellar genes we utilized the IMG/M platform (*Chen et al., 2017*).

### Bacterial motility assay
Motility was assessed by examining the expansion of bacterial colonies plated on semi-solid agar (*Wolfe and Berg, 1989*). Semi-solid media of Exp-CM, Exp-CM +DMSP, and Stat-CM were prepared by mixing boiling sterile 3% agarose with CM, which was pre-heated to ~50°C, in a 1:9 ratio (final concentration of 0.3% agarose). Media was quickly distributed in six-well plates,~5 mL per plate, and was left to solidify for ~1 hr. *Sulfitobacter* D7 was pre-grown in liquid Exp-CM, Exp-CM +DMSP, and Stat-CM in order to induce the appropriate expression of flagellar genes. For control, bacteria were pre-grown in liquid ½MB, lacking algal DMSP and infochemicals. After 24 hr bacterial abundance was

evaluated and the concentration of bacteria in each media was normalized to $2 \cdot 10^6$ mL$^{-1}$, to ensure that the difference in colony size would be indicative of motility and not abundance of bacteria. Bacteria grown in CM were plated on the corresponding semi-solid CM (0.3% agarose), and bacteria grown in ½MB were plated on each semi-solid CM. For plating, 1 µL of bacteria were pipetted in the center of each well containing semi-solid media, in 10–12 replicates per treatment. Colonies were visualized with ×2 magnification after 6 days using Nikon SMZ18 Steriomicroscope. Colonies measurements were performed using the Annotation and Measurements tool of the Nikon NIS-Elements Analysis D software.

## Quantification of benzoate in the media of *Sulfitobacter* D7 mono-cultures

*Sulfitobacter* D7 was inoculated into MM supplemented with 100 µM benzoate in 500 mL volume in triplicates. For control, we also sampled uninoculated medium in triplicates. To quantify extracellular benzoate concentrations, bacterial cultures or uninoculated media were gently filtered, acidified, and led through solid-phase extraction (SPE) cartridges, as described in *Kuhlisch et al., 2021* per sample, 50 mL culture was passed through 0.22 µm polyvinylidene difluoride (PVDF) filters, collected in glass Erlenmeyer flasks, and spiked with 5 µL benzoate-d$_5$ (98%, Cambridge Isotope Laboratories, Tewsbury, MA, USA; 1.276 µg/µL in MeOH) as internal standard (IS; 1 µM final concentration). The filtrates were incubated for 30 min and then acidified to pH 2.0 using 10% HCl. Benzoate was extracted using hydrophilic-lipophilic balanced SPE cartridges (Oasis HLB, 200 mg, Waters, Milford, MA, USA) as follows: cartridges were conditioned (6 mL methanol), equilibrated (6 mL 0.01 N HCl), and loaded by gravity with the acidified samples (45 min). The cartridges were then washed (18 mL 0.01 N HCl), dried completely using a vacuum pump, and gravity-eluted with 2×2 mL methanol into 4 mL glass vials. Eluates were stored at –20°C overnight, dried under a flow of nitrogen at 0.5 mL/min and 30°C (TurboVap LV, Biotage, Uppsala, Sweden), and stored at –20°C until further processing. Glassware and chemically resistant equipment were used whenever possible and cleaned with HCl (1 or 10%) and Deconex 20 NS-x (Borer Chemie, Zuchwil, Switzerland) to reduce contaminations.

For LC-MS analysis, the dried extracts were thawed, re-dissolved in 300 µL methanol:water (1:1, v:v), vortexed, sonicated for 10 min, and centrifuged at 3200 × g for 10 min at 4°C. The supernatants were transferred to 200 µL glass inserts in autosampler vials and analyzed by ultra-high-performance liquid chromatography (UPLC)-electrospray ionization(ESI)-high resolution mass spectrometry. An aliquot of 1 µL was analyzed using UPLC coupled to a photodiode array (PDA) detector (ACQUITY UPLC I-Class, Waters) and a quadrupole time-of-flight mass spectrometer (SYNAPT G2 HDMS, Waters), as described previously (*Kuhlisch et al., 2021*) with slight modifications. Chromatographic separation was carried out using an ACQUITY UPLC BEH C18 column (100×2.1 mm, 1.7 µm; Waters) attached to a VanGuard pre-column (5×2.1 mm, 1.7 µm; Waters). The mobile phase, at a flow rate of 0.3 mL/min, consisted of water (mobile phase A) and acetonitrile (mobile phase B), both with 0.1% formic acid, and set as follows: a liner gradient from 100 to 75% A in 20 min, from 75 to 0% A in 6 min, 2 min of 100% B, and 2 min to return to the initial conditions and re-equilibrate the column. The PDA detector was set to 200–600 nm. A divert valve (Rheodyne) excluded 0–1 min and 25.5–30 min from injection to the mass spectrometer. The ESI source was operated in negative ESI mode (ESI$^-$) and set to 140°C source and 450°C desolvation temperature, 1.0 kV capillary voltage, and 27 eV cone voltage, using nitrogen as desolvation gas (800 L/hr) and cone gas (10 L/hr). The mass spectrometer was operated in full scan MS$^E$ resolution mode with a mass range of 50–1600 Da and the mass resolution tuned to 23,000 at m/z 554 alternating with 0.1 min scan time between low- (4 eV collision energy) and high-energy scan function (collision energy ramp of 15–50 eV).

An external calibration curve was processed in parallel to the biological samples. Aliquots of 100 mL ASW were spiked with 10 µL benzoate-d$_5$ (1 µM final concentration) and benzoate standard solutions to reach final concentrations of 0.2, 1, 2, 10, 20, and 100 µM benzoate. Two blanks were prepared, one blank that was spiked only with the IS, and one blank lacking both IS and benzoate. Each sample was divided to duplicates of 50 mL and extracted as described above. After re-dissolving in 200 µL methanol:water (1:1, v:v), samples were injected subsequent to the biological samples. The peak areas of the [M-H]$^-$ ions for the IS (*m/z* 126.06) and benzoate (*m/z* 121.029) were extracted above a signal-to-noise threshold of 10 using TargetLynx (Version 4.2, Waters). The uniformly labeled aromatic ring of the IS slightly reduces its column retention (*Figure 4—figure supplement 3*). Analyte response

(y) was calculated by dividing the area of benzoate by the IS, plotted against the benzoate concentration (x), and the slope and intercept for a linear regression calculated (y=0.4741 x+1.665, $R^2$=0.99) (*Figure 4—figure supplement 3*). Quantification of benzoate in the biological samples was based on the analyte response in each sample and the calibration curve, with a limit of quantification of 200 nM.

## Quantification of benzoate in the media of *E. huxleyi-Sulfitobacter* D7 co-cultures

*Sulfitobacter* D7 was inoculated to *E. huxleyi* CCMP379 cultures in 440 mL volume. A total of 12 non-infected and 12 infected algal cultures were used, and for each timepoint, 3 replicates of non-infected and infected cultures were sampled entirely for benzoate quantification in the media. Three replicates of algal growth medium (FSW +F/2) were used as medium control. At T=0 days, only three replicates of non-infected cultures were sampled for benzoate quantification. Extracellular benzoate was extracted as described above with slight modifications. Briefly, culture aliquots of 400 mL were filtered gently (<0.4 bar under-pressure) over pre-combusted Whatman glass microfiber GF/A filters to remove algal cells, and subsequently over 0.22 µm PVDF filters to remove bacterial cells. Filtrates were spiked with 5 µL benzoate-$d_5$ (125 nM final concentration), acidified, and metabolites extracted using SPE cartridges. Dried extracts were stored at –80°C until LC-MS analysis. For targeted benzoate analysis, the dried extracts were re-dissolved in 200 µL of 20 mM ammonium carbonate (pH 9.25) in acetonitrile (1:1, v:v) and immediately analyzed by LC-MS/MS. Chromatography was performed on an ACQUITY UPLC I-Class (Waters Corporation); 10 µl of the sample was injected onto a ACQUITY BEH Amide column 2.1 mm × 150 mm, 1.7 µm (Waters) kept at 35°C. A binary gradient was applied using mobile phase A containing 17% acetonitrile in 20 mM ammonium carbonate (pH 9.2), and mobile phase B containing acetonitrile. The gradient elution was performed at 0.3 mL/min with initial inlet conditions at 95% B during 0.5 min, then decreasing to 20% B over 2 min, followed by a column wash at 20% B for 0.5 min, and a return to initial conditions at 95% B over 0.5 min. The total run time was 6 min. A Xevo TQ-S Tandem MS (Waters, Wilmslow, UK) operating in ESI⁻ was used for the detection and quantification of the benzoate using the following instrument conditions: capillary voltage 1.92 kV, source temperature 150°C, desolvation temperature 400°C, cone gas flow 150 L/hr, and desolvation gas flow 700 L/hr. Benzoate was detected using selected reaction monitoring acquisition mode with transitions 121>77 *m/z* for benzoate, 126>82 *m/z* for benzoate-$d_5$, and 11 eV collision energy. The benzoate concentrations based on a 0–100 µM standard curve were calculated as described below.

An external calibration curve was prepared by spiking aliquots of a 1 mM benzoate and 1 mM benzoate-$d_5$ solution into 200 µL re-dissolving solution to reach final concentrations of 0.0125 µM, 0.025 µM, 0.0375 µM, 0.05 µM, 0.0625 µM, 0.075 µM, 0.0875 µM, and 0.1 µM benzoate. Two blanks were prepared as described above. Calibration curve samples were injected prior and subsequent to the biological samples. The peak areas of the fragment ions of the IS (*m/z* 82) and of benzoate (*m/z* 77) were extracted (*Figure 4—figure supplement 3*) using TargetLynx (Version 4.2, Waters). The analyte response was calculated as described above, and the slope for a linear regression calculated (y=0.0072 x, $R^2$=0.99) (*Figure 4—figure supplement 3*). Quantification of benzoate in the medium samples was based on the analyte response in each sample, the calibration curve, and a 0.0005× dilution factor between the injected extract and the culture filtrates.

## Prevalence of benzoate transport and catabolism genes in genomes of phytoplankton-associated bacteria

Bacterial benzoate degradation pathways and the genes encoding for the metabolic enzymes were reconstructed with the use of MetaCyc (*Caspi et al., 2014*) and the KEGG Pathway database (*Kanehisa and Goto, 2000*; *Figure 4—figure supplement 1*). Selected genes, encoding for benzoate transporters and for the enzymes mediating the initial steps of benzoate metabolism in each pathway, were used to search for similar proteins in bacterial genomes using BLASTp. The list of these query genes, which were all previously experimentally validated, is found in the Key resources table. The target bacterial genomes were selected based on their known association with *E. huxleyi* and other phytoplankton species (*Figure 5—source data 1*). Positive hits had an E-value <0.005, identity >30%, and coverage >30. Hits with lower coverage and/or identity were considered as 'Partial'. The results are summarized in *Figure 5—source data 1*.

## Statistical analyses

For the motility assay (*Figure 2b*) and bacterial growth on carbon sources (*Figure 5b*) we used two-way ANOVA, followed by Tukey's post-hoc test, using the R-package 'emmeans'. For benzoate consumption experiments (*Figure 4c and d*) we used a mixed effects model, with treatment and time as fixed effects, and replicate as a random effect, using the R packages 'lme4' and 'lmerTest'. For the *E. huxleyi* and bacterial growth curves (*Figure 5c* and *Figure 6*) we used repeated-measures ANOVA, followed by a Tukey post hoc test, using the R package 'emmeans'. All analyses were done using R, v. 4.1.2.

## Figures preparation

Figures and illustrations were prepared using PowerPoint, Excel, and BioRender.com.

## Acknowledgements

We thank Ron Rotkopf for his assistance in statistical analysis. We thank Daniella Schatz for constructive feedback and scientific discussions. We thank Einat Segev for providing the *Sulfitobacter* strains *S. pontiacus* DSM 10014 and *S. brevis* DSM 11443. We thank Assaf R Gavish for fruitful discussions and assistance in graphics.

## Additional information

### Funding

| Funder | Grant reference number | Author |
|---|---|---|
| European Research Council | 681715 | Assaf Vardi |
| European Research Council | 101053543 | Assaf Vardi |

The funders had no role in study design, data collection and interpretation, or the decision to submit the work for publication.

### Author contributions

Noa Barak-Gavish, Conceptualization, Data curation, Formal analysis, Investigation, Visualization, Methodology, Writing – original draft, Writing – review and editing; Bareket Dassa, Data curation, Formal analysis, Investigation, Writing – review and editing; Constanze Kuhlisch, Formal analysis, Investigation, Methodology, Writing – review and editing; Inbal Nussbaum, Gili Rosenberg, Roi Avraham, Investigation, Writing – review and editing; Alexander Brandis, Methodology, Writing – review and editing; Assaf Vardi, Conceptualization, Funding acquisition, Investigation, Writing – original draft, Project administration, Writing – review and editing

### Author ORCIDs

Noa Barak-Gavish http://orcid.org/0000-0002-3881-884X
Constanze Kuhlisch http://orcid.org/0000-0002-7823-5382
Roi Avraham http://orcid.org/0000-0002-9098-3885
Assaf Vardi http://orcid.org/0000-0002-7079-0234

### Decision letter and Author response

Decision letter https://doi.org/10.7554/eLife.84400.sa1
Author response https://doi.org/10.7554/eLife.84400.sa2

## Additional files

### Supplementary files
• MDAR checklist

## Data availability

The RNAseq data was deposited in NCBI's Gene Expression Omnibus (GEO) and is available through GEO series accession number GSE193203. All data generated or analyzed during this study are included in the manuscript and Figure 1—source data 2.

The following dataset was generated:

| Author(s) | Year | Dataset title | Dataset URL | Database and Identifier |
|---|---|---|---|---|
| Barak-Gavish N, Dassa B, Kuhlisch C, Nussbaum I, Rosenberg G, Avraham R, Vardi A | 2022 | Bacterial lifestyle switch in response to algal metabolites | https://www.ncbi.nlm.nih.gov/geo/query/acc.cgi?acc=GSE193203 | NCBI Gene Expression Omnibus, GSE193203 |

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
