## [Editor Report]

This paper presents important new findings on the mechanism by which a bacterial species switches between co-existence with a bloom-forming phytoplankter to being pathogenic. Specifically, the study identifies algal dimethylsulfoniopropionate (DMSP) as a key chemical component that triggers the bacterial switch. The results are convincing and will be of interesting to scientists interested in inter-kingdom microbial communication including microbiologists and biologists working with algae.

---

## [Decision Letter]

[Editors' note: this paper was reviewed by Review Commons.]

---

## [Author Response]

1. General Statements

In this study, we aimed to unravel the molecular basis for the bacterial lifestyle switch, from coexistence to pathogenicity, using the ecologically important alga-bacterium model system of *Sulfitobacter* D7 and its interaction with the bloom-forming alga *Emiliania huxleyi. E. huxleyi* forms massive blooms in the oceans and thus play a central role in biogeochemical cycling of elements. Therefore, mechanistic understanding of the metabolic exchange that determines the fate of this algal host has profound ecological significance.

In the current work, we demonstrate that bacteria can sense their host’s physiological state by perceiving the bouquet of secreted chemical signals, which eventually leads to the induction of bacterial pathogenicity. By profiling the transcriptional response of the bacterium, we found that the algal metabolite dimethylsulfoniopropionate (DMSP), an important signaling molecule in the marine environment, is essential for the bacterial lifestyle switch. However, the effective signaling of DMSP depended on additional algal signals. In complex environments, where many microbial species are present simultaneously, such a mechanism can ensure specificity, and thus bacteria will invest in altering gene expression and metabolic remodeling only when the right algal partners are present.

We discovered that the metabolic exchange of two key metabolic currencies, DMSP and benzoate, both produced and released by algal cells, regulate the bacterial lifestyle during interactions. Imbalance in their production, determined by the host physiological state, may induce a shift to bacterial pathogenicity towards its algal host, which could eventually contribute to algal bloom demise in natural populations.

Based on the reviewers’ suggestions, we conducted numerous experiments and added the following data that reinforce our conclusions:

1. We now provide a detailed analysis of benzoate production by *E. huxleyi* and its accumulation in mono-cultures and demonstrate the consumption of benzoate by *Sulfitobacter* D7 in co-cultures (new Figure 4d).

2. We added a new figure (new Figure 5) in which we examined two additional Sulfitobacter species that their genomes do not indicate benzoate utilization (new Figure 5a). We show their inability to grow on benzoate as sole carbon source (new Figure 5b) and their coculturing dynamics with *E. huxleyi* (new Figure 5c).

3. We now strengthen our conclusion that benzoate hinders the DMSP-induced bacterial lifestyle switch to pathogenicity by adding evidence from co-cultures of *Sulfitobacter* D7 with two different strains of *E. huxleyi* (CCMP379, new Figure 6; and CCMP 2090, new Figure S5).

4. We added evidence that benzoate, served as a sole carbon source, promotes the growth of *Sulfitobacter* D7 in a dose-dependent manner (Figure S4).

In addition, we improved the statistical analyses, clarified some points raised by the reviewers regarding the methodologies and made the RNAseq data more accessible.

2. Point-by-point description of the revisionsReviewer #1 (Evidence, reproducibility and clarity (Required)):This manuscript seeks to evaluate the influence of algal-derived metabolites on the transcriptomic expression of genes in a Sulfidobacter species. This particular bacteria has been observed to switch between co-existence with E. huxleyi to more of a pathogenic lifestyle. These authors seek to better understand the mechanisms driving this switch. Overall, I felt that this was a very wellwritten paper, and very clearly walked the reader through the findings and discussion of the results. However, I have some things that the authors should address prior to publication.

We thank the reviewer for the encouraging feedback and useful comments. We have now added numerous experiments that strengthen the role of benzoate during *Sulfitobacter* D7*E. huxleyi* interaction. We also clarified some points raised by the reviewer regarding the methodologies. We also improved and clarified the statistical analyses and significance of our results.

1. I did not think that the authors clearly demonstrated why they decided to focus on the influence of benzoate to this relationship outside of other potential metabolites. The authors state that benzoate degradation genes were highly expressed, but don't show that explicitly. It would be fine as a figure in the supplemental, but just that these genes are more highly expressed relative to others.Additionally, Figure 4b does not show any significance. It is clear that for some genes, the ExpCM+DMSP and Stat-CM treatments are elevated relative to other treatments, however there is no indication if that elevation is significant or not. Denoting that would reinforce the importance of benzoate in mediating this interaction.

We agree with the reviewer. We now added information on the statistical significance in expression values between the CM treatments and the MM treatments. In Figure 4b, we added * which mark genes that were significantly DE between all CM and MM treatments (Exp-CM vs MM, Exp-CM vs MM+DMSP, Exp-CM+DMSP vs MM, Exp-CM+DMSP vs MM+DMSP, Stat-CM vs MM and Stat-CM vs MM+DMSP). Genes marked with ** were significantly DE between all CM and MM treatments but one comparison (for *BenD* Exp-CM vs MM+DMSP; for *PcaF* Exp-CM+DMSP vs MM).

Furthermore, in Supplementary Table S5, the reviewer can find the expression values of benzoate-related genes and which of these were significantly DE.

While aromatic compound degradation is a known feature of Roseobacters, the direct degradation of benzoate seems to be limited to only a few phytoplankton-associated bacteria, based on our analysis (Figure 5a). We now added an experiment that demonstrates that *Sulfitobacter* D7 consumes benzoate during co-culturing, while it is accumulated in algal mono-cultures (Figure 4d). We also demonstrate now that Sulfitobacters that don’t consume benzoate do not exhibit a lifestyle switch (Figure 5b,c), suggesting that the metabolic exchange of benzoate is important for the bacterial lifestyle switch.

In order to further back up our conclusions about the importance of benzoate, we now included results of *Sulfitobacter* D7 co-culturing with two *E. huxleyi* strains (CCMP2090 and CCMP379) (new Figure 6 and Figure S5). Both experiments demonstrate that benzoate hinders the pathogenicity-inducing effect of DMSP, further supporting our conclusions.

We feel that the additional experiments and data we provide in the revised manuscript reinforce our conclusions regarding the involvement of benzoate in *Sulfitobacter* D7-*E. huxleyi* interaction.

2. Were the strains of Ehux used for these experiments axenic? This actually would be a huge factor in influencing the results of this paper and it is not expressly stated in the paper. This needs to be stated, and if they were not axenic, the authors need to discuss how this may influence the results observed.

The algal cultures were axenic we routinely checked for bacterial contamination in the algal cultures by counting bacteria in algal cultures by flow cytometry. We now added a statement regarding this in the Materials and methods (L432-434).

3. The DMSP concentrations used in these experiments are higher than what has been measured previously being exuded from bacteria. Are the authors assuming that this would be a concentration bacteria would observe in the phycosphere? If so, that should be explicitly written somewhere in the manuscript. Similarly, for benzoate.

The reviewer is correct, the higher concentrations used in our study aimed to mimic concentrations that bacteria would be exposed at the surface of phytoplankton cells (in the phycosphere). We have now clarified this point in the method section, which now reads: “This concentration mimics that present in the phycosphere of *E. huxleyi* in stationary phase^1,2^.” (L457-L458)

4. Minor methods questions: Figure S1 – Why is there always one replicate that is not aligned on the tree or have as much similarity to the other two replicates in each treatment? Is this a function of how this dendrogram was created? Or is there that much variability among the replicates?

The structure of the dendrogram is created such that it draws a hierarchical binary tree, and therefore one of the triplicates is drawn on a different branch. Still, the height of the branches reflects the distance between the replicates, and one can observe that overall, the triplicates are similar. This is also in agreement with the correlation values, which are very high among the replicates.

What is the nominal pore size of a GF/C filter, and on line 432 "subsequently filtered through 0.22 μm of what kind of filter?" Same issue with line 529 – what kind of 0.22 μm filters?

The pore size of GF/C is 1.2 µm. We now indicated what kind of filters we used for each filtration (L453, L455)

5. Why not use a repeated measures ANOVA for bacterial and algal growth curves? You are repeatedly measuring cell abundance from single flasks. The repeated measures ANOVA would disentangle the change in growth within a treatment over time, with differences in growth between treatments. Further, the mixed effects models require the data to be linear. Was only a fraction of the growth curve used to look for significant differences? If not, the growth curve isn't linear, therefore doesn't meet the assumptions of the mixed effects models.

We now changed the statistical test for the algal and bacterial growth curves to repeated measures ANOVA in new Figure 6 and Figure S5, and indicated this in the “statistical analyses” section (L659).

Reviewer #1 (Significance (Required)):This publication advances our understanding of the mechanisms that drive algal-bacteria interactions. It will be of interest to microbial ecologists as well as those interested in the chemical ecology of microbes.

We thank the reviewer for acknowledging the quality and scope of our work.

Reviewer #2 (Evidence, reproducibility and clarity (Required)):SummarySulfitobacter D7 changes its lifestyle from coexistence to pathogenicity when interacting with E. huxleyi. During the coexistence phase, E. huxleyi secretes metabolites such as benzoate, DMSP, and other growth substrates into the phycosphere, which bacteria can absorb and use for growth. The authors postulated that such energy-rich metabolic currencies inhibit DMSP signalling in Sulfitobacter D7 based on the discovery that benzoate neutralized the pathogenicity-inducing impact of DMSP. This observation is in contrast or complementary to previous findings. The amount of accessible growth substrates reduces when the algal physiological state is disturbed, such as during stationary growth, due to bacterial consumption and reduced secretion by the alga. In this study, a very high applied concentration of algal DMSP functions as a signal, altering the bacterium's transcriptional patterns and increasing the expression of pathogenicity-related genes.Lifestyle switches were often reported but rarely reproduced by the scientific community. The paper is very well and interestingly written and illustrated. Further experiments should back up the far-reaching conclusions to ensure reproducibility.

We thank the reviewer for acknowledging the quality and scope of our work. In order to strengthen the conclusions of our work, we conducted numerous experiments that the reviewer requested:

1. We now provide a detailed analysis of benzoate production by *E. huxleyi* and its accumulation in mono-cultures and demonstrate the consumption of benzoate by *Sulfitobacter* D7 in co-cultures (new Figure 4d).

2. We added a new figure (new Figure 5) in which we examined two additional Sulfitobacter species that their genomes do not indicate benzoate utilization (new Figure 5a). We show their inability to grow on benzoate as sole carbon source (new Figure 5b) and their co-culturing dynamics with *E. huxleyi* (new Figure 5c).

3. We now strengthen our conclusion that benzoate hinders the DMSP-induced bacterial lifestyle switch to pathogenicity by adding evidence from co-cultures of *Sulfitobacter* D7 with two different strains of *E. huxleyi* (CCMP379, new Figure 6; and CCMP 2090, new Figure S5).

4. We added evidence that benzoate, served as a sole carbon source, promotes the growth of Sulfitobacter D7 in a dose-dependent manner (new Figure S4)Major comments– Very high concentrations of DMSP (100 µM) were applied for transcriptional profiling (line 433, line 326, Figure5)? In specific experimental approaches, such high concentrations may be justified but not for a stimulus in transcriptome analysis. This calls into question the ecological relevance of the study.

We respectfully disagree with the reviewer on this point. Although the concentration of DMSP measured in seawater are typically in the tens of nanomolar range^3^, these compounds are not homogeneously distributed in the water column^4^. Indeed, phytoplankton internal concentrations of DMSP can exceed 100 mM ^5,6^ , therefore DMSP concentrations will be orders of magnitude higher than background within the microenvironments surrounding phytoplankton (phycosphere)^1^ or as a result of cell damage or lysis^7^. Furthermore, the concentrations of DMSP measured in axenic stationary phase *E. huxleyi* cultures in our previous studies reach up to 70 µM^2^. We therefore argue that the concentrations used in our study are appropriate and ecologically relevant.

– Do you know the natural concentration of DMSP and benzoate in the growth medium of your laboratory cultures? Did you also test lower concentrations than 100 µM?

The concentration of DMSP in the media of axenic *E. huxleyi* cultures ranges between 0.5 to 70 µM, depending on the growth phase and algal strain. This data is presented in our previous study ^2^. We now added additional results showing the concentration of benzoate in the media of algal mono-cultures and during co-culturing with *Sulfitobacter* D7 (Figure 4d). The concentration of benzoate in the algal mono-culture’s medium was in the tens of nM range. Nevertheless, the concentration in the phycosphere is likely much higher^1^. We did test lower concentrations then 100 µM. Please see Author response image 1 related to the DMSP/benzoate ratios.

**Author response image 1. sa2fig1:** Time course of *E. huxleyi* CCMP379 mono-cultures (dashed lines) and during co-culturing with *Sulfitobacter* D7 (smooth lines). Cultures we either not supplemented (a-b) or supplemented with 100 µM DMSP at day 0 (c-d). Benzoate was added to the cultures at day 0 in the following concentrations: 1 µM (blue), 10 µM (green), 100 µM (yellow) or none (grey).

– Please provide the EC50 value of the benzoate? That information is essential to understanding the bacteria-algae interactions.

We now added results showing the dose-dependent growth promoting effect of benzoate on *Sulfitobacter* D7 (Figure S4). We also performed a co-culturing experiment in which we added different concentrations of benzoate, with and without 100 µM DMSP. From these results, equimolar concentrations of 100 µM benzoate seems to be most effective for bacterial growth and for hindering the pathogenicity-inducing DMSP signaling.

– In general, I recommend performing knockout mutants of Sulfitobacter D7 (e.g,, benzoate degradation) to evaluate further the DMSP/benzoate regulated association with E. huxleyi. It will certainly take a while, but supporting such a far-reaching study and its conclusions is worthwhile.

We agree with the reviewer that this approach will be beneficial and help support our claims. However, unfortunately generating *Sulfitobacter* D7 knockout strains is currently not feasible. *Sulfitobacter* D7 was isolated by us from the natural environment^2^. Although we established a meticulous and robust model system, this bacterium is not genetically amenable. We invested strong efforts on genetically modifying *Sulfitobacter* D7, but still with no successes. Therefore, this request is beyond the scope of the current manuscript, and we are currently working towards this goal.

– A control experiment with a closely related non-benzoate degrading strain of Sultfitobacter spp. (listed in Figure 4d) would be desirable (Figure 4c). Is it possible that the depletion of benzoate in Figure 4c is due to adsorption by the cells? Also, I do not see statistical differences in the treatments shown in Figure 4b.

We now added a new figure (Figure 5) in which we show the growth on benzoate as sole carbon source and the co-culturing dynamics of two additional Sulfitobacter strains that cannot utilize benzoate. We show that *S. pontiacus* DSM 10014 and *S. brevis* DSM 11443 do not have the genes required for benzoate utilization (Figure 5a). Accordingly, these strains do not grow on benzoate as sole carbon source (Figure 5b). We also show that when we cocultured *S. pontiacus* and *S. brevis* with *E. huxleyi* they do not switch their lifestyle to pathogenicity (Figure 5c). We further discuss this observation in the manuscript.

Regarding the depletion of benzoate in Figure 4c- we are confident that it is derived by bacterial consumption and metabolism and not due to adsorption by the cells. In this experiment *Sulfitobacter* D7 grew 4-orders of magnitude, therefore the bacteria must have consumed and metabolized benzoate to gain energy for growth.

Regarding the statistical differences in Figure 4b- we now added information on the statistical significance in expression values between the CM treatments and the MM treatments. In Figure 4b, we added * which mark genes that were significantly DE between all CM and MM treatments (Exp-CM vs MM, Exp-CM vs MM+DMSP, Exp-CM+DMSP vs MM, Exp-CM+DMSP vs MM+DMSP, Stat-CM vs MM and Stat-CM vs MM+DMSP). Genes marked with ** were significantly DE between all CM and MM treatments but one comparison (for *BenD* Exp-CM vs MM+DMSP; for *PcaF* Exp-CM+DMSP vs MM).

Furthermore, in Supplementary Table S5, the reviewer can find the expression values of benzoate-related genes and which of these were significantly DE.

– Did you prove the biosynthetic pathway by qPCR analysis? The benzoate degradation could be determined using labelled benzoate.

We now added new results that show the algal production of benzoate and consumption of benzoate by *Sulfitobacter* D7 during co-culturing with *E. huxleyi* (Figure 4d). We invested tremendous efforts to achieve these results, including extensive LC-MS method development and analysis. Furthermore, our results from the transcriptomics experiment demonstrate the co-transcription of the whole pathway in CM indicating that the whole pathway is indeed active if benzoate is present (Figure 4b, Table S5). This, together with our previously presented results of benzoate consumption in bacterial mono-cultures (Figure 4c), clearly demonstrates that *Sulfitobacter* D7 uptakes benzoate.

– The lifestyle switch appears to be regulated by a certain DMSP/benzoate ratio. The use of such a ratio as a proof of principle would be extremely beneficial and required to support the assumptions and results which are given in Figure 6.

We performed an experiment in which we added different concentrations of benzoate (1, 10 and 100 µM) with and without 100 µM DMSP, to see if there is a dose-dependent effect. While the 100 µM benzoate hinders DMSP signaling statistically significant, the 10 µM benzoate concentration shows some effect, though not statistically significant (see Author response image 1).

In order to back up our conclusions, we now include results of *Sulfitobacter* D7 coculturing with two *E. huxleyi* strains (CCMP2090 and CCMP379) (new Figure 6 and Figure S5). Both experiments demonstrate that benzoate hinders the pathogenicity-inducing effect of DMSP, further supporting our conclusions.

– I am curious as to what the author thinks is going on, if the second source of benzoate (other algae) is present? How equal are all the E. huxleyi cells within a bloom (single-cell analysis and quantification of benzoate release might be necessary)?

The reviewer raises an interesting point. Taking into consideration the scale at which interactions occur, we think that the interaction depends on the composition of each phycosphere. In a diluted environment, such as the marine environment, bacteria will most likely not sense benzoate from a different alga, unless a bacterium is within the algal phycosphere. In lab cultures the situation is different, because there is no mixing and molecules can accumulate in the cultures’ media. Therefore, an additional source of benzoate can indeed alter the interaction with the algae, as we see in our observations (Figure 6 and Figure S5).

The second question the reviewer raised is very difficult to answer. Work on the heterogeneity of algal cells within a bloom is technologically challenging. Nevertheless, work from our group using single-cell transcriptomics, showed that during viral infection of *E. huxleyi* cultures the algal population is quite heterogeneous, even in isogenic populations^8^. Regarding the metabolic activity at the single-cell level, although there have been some advances in recent years^9,10^, this would still be extremely challenging to conduct and will required method development of several years. We think this subject is very interesting but beyond the scope of the current study.

Minor comments– Line 344: Is the statement supported by a representative study?

The analysis that we performed and presented in new Figure 5a supports this.

– Line 378: What does "eat-and-run" strategy mean?

It refers to the English idiom “eat and run” which describes a situation where someone eats quickly and leaves.

– If I understand the method section correctly, you have worked with two organisms only, Sulfitobacter D7 and E. huxleyi. How did you prove that no bacterial contaminants were influencing the experiment?

The algal cultures were axenic, and we routinely checked for bacterial contamination in the algal cultures by counting bacteria in algal cultures by flow cytometry. We now added a statement regarding this in the Materials and methods (L432-434).

– DMSP is produced and released by multiple organisms. It possesses multiple modes of action depending on the assay performed. A more general introduction and discussion would be desirable. The same is true for benzoate.

Regarding DMSP, we address this point in the introduction (L52-55):

“The organosulfur molecule dimethylsulfoniopropionate (DMSP), produced by many phytoplankton species^11^, is especially known to mediate Roseobacter-phytoplankton interactions by serving as a carbon and sulfur source, a chemotaxis cue and as an infochemical for the presence of algae^2,7,12–17^.” And in the discussion (L336-338):

“DMSP is a ubiquitous infochemical produced by many phytoplankton species as well as some bacteria^18^, making it a prevalent signaling molecule that mediates microbial interactions in the marine environment….”

Regarding benzoate, to the best of our knowledge, there is almost no literature about benzoate in the marine environment. There is, however, literature on aromatic compound degradation by Roseobacters, and we discuss this in L272-274. There are also no reports that show directly benzoate production by phytoplankton. This was, in part, what motivated us to measure benzoate in *E. huxleyi* cultures (Figure 4d), and to explore the prevalence of benzoate-related genes in phytoplankton-associated bacteria (Figure 5a).

Why did you use different strains of E. huxleyi in experiments presented in Figure 1 (transcriptomics) and Figure 5 (benzoate experiment)?

We now added a similar experiment with *E. huxleyi* strain 379, which is the strain presented in Figure 1. This data is found in new Figure 6. We moved the results with *E. huxleyi* strain 2090 to the supplementary (Figure S5). The difference between the two strains is the ability to naturally induce the pathogenicity of *Sulfitobacter* D7. This is attributed to the difference in DMSP concentration between the strains. The concentration of DMSP in the strain presented in Figure 1 (*E. huxleyi* 379) in higher by 10-fold than the concentration in the strain presented in the new Figure S5 (*E. huxleyi* 2090)^2^. The reason we used 2090 in the experiment in new Figure S5 is that we wanted to examine how the DMSP-benzoate interplay affects the bacterial lifestyle switch, so we used the strain in which we can induce the switch.

Reviewer #2 (Significance (Required))Significance.The study is organized excitingly and characterized by various experiments that support each other but are not yet final for the far-reaching conclusions. In particular, the ecological relevance is not yet given and direct proofs for the conclusions made under laboratory conditions are missing. With additional experiments, the authors can validate their interesting observations and place them in an ecologically relevant context, which further increases the significance of the study.

We thank the reviewer for the encouraging and constructive feedback on our work. We feel that the new data that we added, based on the reviewer’s suggestions, indeed strengthen our conclusions.

Reviewer #3 (Evidence, reproducibility and clarity (Required)):Summary:In the current manuscript the authors expand on their previous finding that DMSP acts as a trigger for Paracoccus D7 to become pathogenic towards the microalgae Emiliania huxleyi in co-culture. They performed a series of transcriptome experiments in co-culture-derived and minimal medium with or without externally added DMSP that demonstrated that DMSP acts as an infochemical leading to major changes in the D7 transcriptome when other algae-derived metabolites are present, but not in a minimal medium. In particular, they found an activation of the flagella gene cluster and various genes for transporters. Furthermore, they found that the benzoatedegradation pathway is upregulated only in the co-culture-derived medium and that benzoate likely counteracts the signalling role of DMSP.Major comments:The authors performed a carefully planned transcriptome study and tested the hypothesis generated from its outcome in well-designed physiological experiments. I have no major concerns regarding the experimental design and interpretation of the results. Although the authors provide a table with the RNAseq data, I think some useful annotation information is missing here that could be easily added and will enhance the reusability of these data: The column with Locus tag contains not only Locus Tags but also gene symbols for some of the genes. In a few cases (lldD, gatA/B, gabD, some ncRNAs and ssrA (tmRNA)), the Locus tag is missing completely. This should be corrected. Furthermore, Paracoccus D7 harbours several plasmids, however the replicon information is missing. I suggest to include columns with the accession number of the replicon, start, end and strand to the annotation data.

We thank the reviewer for the encouraging and constructive feedback on our work. We now corrected and modified the table of the RNAseq data (Table S2). The table now includes all the correct locus tags, the localization of each gene (chromosome or plasmid), replicon start and end and the strand.

Minor comments:The authors used the term "stationary growth" throughout the manuscript. I would use stationary phase instead, as stationary means no growth.

We now corrected to “stationary phase” throughout the manuscript.

P3 L52: I recommend using the term Roseobacter group for marine Rhodobacteraceae as suggested in https://doi.org/10.1038/ismej.2016.198

We now corrected this term and added this reference.

P3 L59: "Roseobacters often produce essential B-vitamins" Please add https://doi.org/10.1038/ismej.2009.94 demonstrating that D. shibae provides B1 and B12 to two species of microalgae

Done.

P5 L131: "Namely DE in…" also in Stat-CM vs Exp-CM?

Yes.

P6 L165: "operon-like structure" This term is not appropriate. The flagella genes are localized in a gene cluster consisting of several (putative) operons. As the authors have sequenced the transcriptome in paired-end mode, I think they could even identify operons from these data. This is however beyond the scope of this manuscript (but might be interesting in general). I just would use the term flagella gene cluster here.

We now corrected the “operon-like” term to “gene cluster” is all relevant places in the manuscript.

P8 L227: for each medium or for each of the media

We corrected this “…for each medium”.

P9 L254: again, I think "operon-like structure" is not the correct term.

We changed to “gene cluster”.

P12 L349: that select.

Corrected.

Reviewer #3 (Significance (Required)):In recent years it became evident that the interaction of several species of the Roseobacter group with microalgae hosts is characterized by a switch from mutualism to pathogenicity. Here, the authors expand on the current knowledge by demonstrating that this switch is probably controlled by a complex interplay between several algal-derived metabolites. These findings are of great interest to all researches in the fields of marine microbiology and microbial symbiosis Own expertise: Microbiology, with main focus on genomics and transcriptomics as well as infochemicals (Quorum sensing) and bacteria-host interactions.

We thank the reviewer for acknowledging the quality and scope of our work.

References

Seymour, J. R., Amin, S. A., Raina, J.-B. & Stocker, R. Zooming in on the phycosphere: the ecological interface for phytoplankton–bacteria relationships. *Nat Microbiol* 2, 17065 (2017).Barak-Gavish, N. *et al.* Bacterial virulence against an oceanic bloom-forming phytoplankter is mediated by algal DMSP. *Sci Adv* 4, eaau5716 (2018).Kettle, A. J. *et al.* A global database of sea surface dimethylsulfide (DMS) measurements and a procedure to predict sea surface DMS as a function of latitude, longitude, and month. *Global Biogeochem Cycles* 13, 399–444 (1999).Stocker, R. Marine Microbes See a Sea of Gradients. *Science (1979)* 338, 628–633 (2012).Stefels, J., Steinke, M., Turner, S., Malin, G. & Belviso, S. Environmental constraints on the production and removal of the climatically active gas dimethylsulphide (DMS) and implications for ecosystem modelling. *Phaeocystis, Major Link in the Biogeochemical Cycling of Climate-Relevant Elements* 245–275 (2007) doi:10.1007/978-1-4020-62148_18.Steinke, M., Wolfe, G. V. & Kirst, G. O. Partial characterisation of dimethylsulfoniopropionate (DMSP) lyase isozymes in 6 strains of *Emiliania huxleyi*. *Mar Ecol Prog Ser* 175, 215–225 (1998).Seymour, J. R., Simó, R., Ahmed, T. & Stocker, R. Chemoattraction to dimethylsulfoniopropionate throughout the marine microbial food web. *Science* 329, 342–345 (2010).Ku, C. *et al.* A single-cell view on alga-virus interactions reveals sequential transcriptional programs and infection states. *Sci Adv* 6, eaba4137 (2020).Behrendt, L. *et al.* PhenoChip: A single-cell phenomic platform for high-throughput photophysiological analyses of microalgae. *Sci Adv* 6, (2020).Alcolombri, U., Pioli, R., Stocker, R. & Berry, D. Single-cell stable isotope probing in microbial ecology. *ISME Communications* 2, 55 (2022).Keller, M. D. Dimethyl Sulfide Production and Marine Phytoplankton: The Importance of Species Composition and Cell Size. *Biological Oceanography* 6, 375–382 (1989).Miller, T. R. & Belas, R. Dimethylsulfoniopropionate metabolism by *Pfiesteria*-associated *Roseobacter* spp. *Appl Environ Microbiol* 70, 3383–3391 (2004).Miller, T. R., Hnilicka, K., Dziedzic, A., Desplats, P. & Belas, R. Chemotaxis of *Silicibacter* sp. strain TM1040 toward dinoflagellate products. *Appl Environ Microbiol* 70, 4692–4701 (2004).

14. Landa, M., Burns, A. S., Roth, S. J. & Moran, M. A. Bacterial transcriptome remodeling during sequential co-culture with a marine dinoflagellate and diatom. *ISME Journal* 11, 2677–2690 (2017).

15. Amin, S. A. *et al.* Interaction and signalling between a cosmopolitan phytoplankton and associated bacteria. *Nature* 522, 98–101 (2015).

16. Sule, P. & Belas, R. A novel inducer of Roseobacter motility is also a disruptor of algal symbiosis. *J Bacteriol* 195, 637–46 (2013).

17. Bürgmann, H. *et al.* Transcriptional response of *Silicibacter pomeroyi* DSS-3 to dimethylsulfoniopropionate (DMSP). *Environ Microbiol* 9, 2742–2755 (2007).

18. Curson, A. R. J. *et al.* Dimethylsulfoniopropionate biosynthesis in marine bacteria and identification of the key gene in this process. *Nat Microbiol*
**2**, 17009 (2017).